# Integrative single-cell characterization of a frugivorous and an insectivorous bat kidney and pancreas

Wei E. Gordon [1,2,13,14], Seungbyn Baek [3,14], Hai P. Nguyen[1,2], Yien-Ming Kuo[4], Rachael Bradley[1,2], Sarah L. Fong [1,2], Nayeon Kim[3], Alex Galazyuk [5], Insuk Lee [3,6], Melissa R. Ingala[7], Nancy B. Simmons [8], Tony Schountz [9], Lisa Noelle Cooper [10], Ilias Georgakopoulos-Soares [11], Martin Hemberg [12] ✉ & Nadav Ahituv [1,2] ✉

Frugivory evolved multiple times in mammals, including bats. However, the cellular and molecular components driving it remain largely unknown. Here, we use integrative single-cell sequencing (scRNA-seq and scATAC-seq) on insectivorous (*Eptesicus fuscus*; big brown bat) and frugivorous (*Artibeus jamaicensis*; Jamaican fruit bat) bat kidneys and pancreases and identify key cell population, gene expression and regulatory differences associated with the Jamaican fruit bat that also relate to human disease, particularly diabetes. We find a decrease in loop of Henle and an increase in collecting duct cells, and differentially active genes and regulatory elements involved in fluid and electrolyte balance in the Jamaican fruit bat kidney. The Jamaican fruit bat pancreas shows an increase in endocrine and a decrease in exocrine cells, and differences in genes and regulatory elements involved in insulin regulation. We also find that these frugivorous bats share several molecular characteristics with human diabetes. Combined, our work provides insights from a frugivorous mammal that could be leveraged for therapeutic purposes.

Frugivory, the ability to thrive on a mostly fruit diet, has allowed mammals to expand their ecological niches[1]. In bats, frugivory has evolved independently from the insectivorous ancestor[2] in two separate families: Pteropodidae (Old World fruit bats; OWFBs) and Phyllostomidae (New World fruit bats; NWFBs)[3]. Adaptation to a primarily fruit-rich diet requires multiple morphological and metabolic adaptations. In bats, these include sensory adaptations like enhanced vision[4,5] and olfaction[6,7], morphological adaptations including the

[1]Department of Bioengineering and Therapeutic Sciences, University of California, San Francisco, San Francisco, CA 94158, USA. [2]Institute for Human Genetics, University of California, San Francisco, San Francisco, CA 94158, USA. [3]Department of Biotechnology, College of Life Science and Biotechnology, Yonsei University, Seoul 03722, Republic of Korea. [4]Department of Ophthalmology, University of California, San Francisco, San Francisco, CA 94143, USA. [5]Hearing Research Focus Area, Department of Anatomy and Neurobiology, Northeast Ohio Medical University, Rootstown, OH, USA. [6]POSTECH Biotech Center, Pohang University of Science and Technology (POSTECH), Pohang 37673, Republic of Korea. [7]Department of Biological Sciences, Fairleigh Dickinson University, Madison, NJ 07940, USA. [8]Division of Vertebrate Zoology, Department of Mammalogy, American Museum of Natural History, New York, NY 10024, USA. [9]Department of Microbiology, Immunology, and Pathology, College of Veterinary Medicine and Biomedical Sciences, Colorado State University, Fort Collins, CO 80523, USA. [10]Musculoskeletal Research Focus Area, Department of Anatomy and Neurobiology, Northeast Ohio Medical University, Rootstown, OH 44272, USA. [11]Institute for Personalized Medicine, Department of Biochemistry and Molecular Biology, The Pennsylvania State University College of Medicine, Hershey, PA 17033, USA. [12]Gene Lay Institute of Immunology and Inflammation, Brigham and Women's Hospital and Harvard Medical School, Boston, MA 02115, USA. [13]Present address: Department of Biology, Menlo College, 1000 El Camino Real, Atherton, CA 94027, USA. [14]These authors contributed equally: Wei E. Gordon, Seungbyn Baek. ✉e-mail: mhemberg@bwh.harvard.edu; nadav.ahituv@ucsf.edu

cranium[8-12], tongue[13], small intestine[14,15], kidney[16,17], and pancreas[18,19], and metabolic adaptations including transporter losses[20,21] and increased transporter activity[3,16].

One of the main challenges a fruit-centered diet poses is high blood sugar levels, which can lead to diabetes. To overcome this, an animal must rapidly control high sugar levels. Interestingly, although fruit bats consume more sugar than non-frugivorous bats, they can lower their blood sugar faster[3,22-24]. Fruit bats demonstrate high sensitivity to glucose and insulin[19] and can directly fuel their metabolic needs with exogenous sugars[25]. The pancreas is responsible for generating hormones that regulate blood sugar and appetite, such as insulin and glucagon[26], as well as secreting enzymes and digestive juices into the small intestine[27]. The high sensitivity to sugar levels in fruit bats is thought to be supported by an expansion of endocrine tissue in the pancreas[18,19,28] and by the loss of genes involved in insulin metabolism and signaling[20].

Fruit is rich in water and low in electrolytes like sodium and calcium. The kidney is responsible for maintaining water and salt balance, filtering the blood of waste, and maintaining blood pressure[29]. The kidney is also involved in metabolizing sugars, generating glucose, and clearing insulin from circulation[30]. Several kidney modifications evolved in fruit bats that lead to dilute urine production[31], including an increased renal cortex and a decreased renal medulla relative to insect-eating bats[16,17] as well as several transporter losses[20].

How mammals adapted to new food sources at the molecular level remains largely unknown. Genetic investigations of insectivory have identified evolutionary adaptations of metabolic enzymes, including gene duplication events of chitin-hydrolyzing enzymes[32,33]. Comparative investigations of mammalian frugivorous adaptation have been primarily performed on a gene-by-gene basis, focusing on metabolism. These include genes such as *GYS1* and *GYS2* (glycogenesis), *NRF2* (antioxidant regulation), *TAT* (protein catabolism), *SLC2A4* (glucose transport), and *AGT* (glyoxylate detoxification)[34-40]. In terms of gene regulatory elements, there are a limited number of studies. One example, is an 11 base pair (bp) deletion observed in both fruit bat families (OWFB and NWFB) in the proximal promoter of *SLC2A2*, which encodes the glucose transporter 2 (GLUT2), that was suggested to be responsible for the difference in liver *SLC2A2* expression in fruit bats[3]. However, no systematic unbiased large-scale genomic studies have been performed to comprehensively identify the cellular and molecular factors governing frugivory adaptation.

Here, we used integrative single-cell RNA-seq and ATAC-seq on adult insectivorous (*Eptesicus fuscus*; big brown bat) and frugivorous (*Artibeus jamaicensis*; Jamaican fruit bat) bats to identify cell populations, genes, and regulatory element differences that could potentially be associated with frugivory adaptations in the kidney and pancreas. In total, we analyzed over 34,696 cells from eight big brown bats and seven Jamaican fruit bats. We developed a cross-species integrated analysis framework for non-model organism genomes, and utilized human and mouse single-cell kidney and pancreas markers to annotate cell-types. Our data reveal cell-type, gene expression, and gene regulation differences between an insectivorous and frugivorous bat. These include more collecting duct cells and gene expression changes involved in fluid and electrolyte balance in the Jamaican fruit bat kidney, and a relative decrease in exocrine cells and gene expression changes involved in insulin secretion and glucose response in the Jamaican fruit bat pancreas. Cell composition differences were further validated with immunofluorescence on insectivorous and frugivorous bat tissues, confirming a reduction in the renal medulla and an expansion of endocrine tissue in Jamaican fruit bats. Transcription factor (TF) analyses of single-cell ATAC-seq found divergent TF binding site (TFBS) usage between dietary phenotypes. Together, our single-cell multi-omics approach indicates that the Jamaican fruit bat kidneys and pancreases exhibit many signatures of diabetes, such as increased potassium secretion, gluconeogenesis, and glucose

reabsorption in the kidney and hyperinsulinemia and hyperglycemia. In summary, our work provides joint single-cell datasets for bats that also compare kidneys and pancreases between closely related mammals of contrasting diets, providing insight into cell composition, gene expression, and gene regulation differences that are also related to human disease, in particular with diabetes.

## Results

### Multiomic single-cell profiling of bat kidney and pancreas

We conducted integrative single-cell sequencing (RNA-seq and ATAC-seq) on the kidneys and pancreases of four adult male insectivorous big brown bats (*Eptesicus fuscus*) (family: Vespertilionidae) and four adult male Jamaican fruit bats (*Artibeus jamaicensis*) (family: Phyllostomidae) to characterize the cell-types, genes and regulatory elements that differ between the two species. The families of these bats diverged approximately 53.8 million years ago (Fig. 1A)[41]. We used these bat species as they have publicly available genomes and are colonized in research labs. Big brown bats were fed a regular diet of mealworms in captivity, whereas Jamaican fruit bats were fed a variety of non-citrus fruits in captivity, such as cantaloupe and banana. We subjected these bats to an overnight fasting regime, followed by two big brown bats fed fruit-fed mealworms (to maximize fruit content) and two fruit bats fed fruit thirty minutes before euthanasia (Fig. 1B). We chose this time point because fruit bats digest food quickly and pass material within 30 minutes[42] and lower their blood sugar within 30 minutes[3,22-24]. Tissues were harvested immediately and flash-frozen. Nuclei were then isolated, subjected to fluorescence-activated cell sorting (FACS), and processed using the 10X Genomics Chromium single-cell Multiome ATAC + Gene Expression kit following established protocols (see Methods).

Because these bats are not widely used model organisms and have poorly assembled and annotated genomes, we made several modifications to analyze their multimodal data. These include (see Methods for more detail): 1) Removal of scaffolds <50 kilo bases (kb) in length. Removal of these short scaffolds still allowed us to capture > 90% of the total sequences and genes for each genome (Supplementary Fig. 1a). 2) Collapse of gene information in annotation files so that each gene is represented by a single "exon" transcript. 3) Due to technical reasons, ATAC-seq can contain a large number of mitochondrial reads[43], whose removal is needed for scATAC-seq analyses. While the Jamaican fruit bat has an annotated mitochondrial genome, the big brown bat does not. We thus used GetOrganelle[44] for assembly and MITOS (Bernt et al. 2013) for annotation to generate a big brown bat mitochondrial genome. Reassuringly, all mitochondrial genes identified in the Jamaican fruit bat genome were detected in the big brown bat genome. 4) Genes often have multiple names, and names can vary across species annotations. Differing gene names affect integration by lowering the number of features shared between species. To improve the integration of scRNA-seq and scATAC-seq between the two bat species, we used Orthofinder v2.5.4[45] to detect one-to-one orthologues, increasing the number of shared features by 3.11% across the big brown bat genome and by 3.24% across the Jamaican fruit bat genome.

We used cellranger-arc 2.0 (10X Genomics) to generate raw ATAC and RNA counts and Seurat[46] and Signac[47] for quality control and downstream analysis (Supplementary Fig. 1b–d, Supplementary Data 1). To merge samples, we created a common peak set across every sample within a species and within a tissue (Supplementary Fig. 2). To increase our sample sizes for phenotypic comparisons between species, we merged all samples within a species, hereinafter referred to as replicates. We combined two fasted and two fed big brown bat samples for each tissue (8527 cells for kidney and 7213 cells for pancreas), two fasted and two fed Jamaican fruit bat samples for the kidney (9315 cells), and two fasted and one fed fruit bat sample (due to one sample providing low quality sequencing data) for the pancreas (9641 cells). To jointly analyze scRNA-seq and scATAC-seq within a species for each

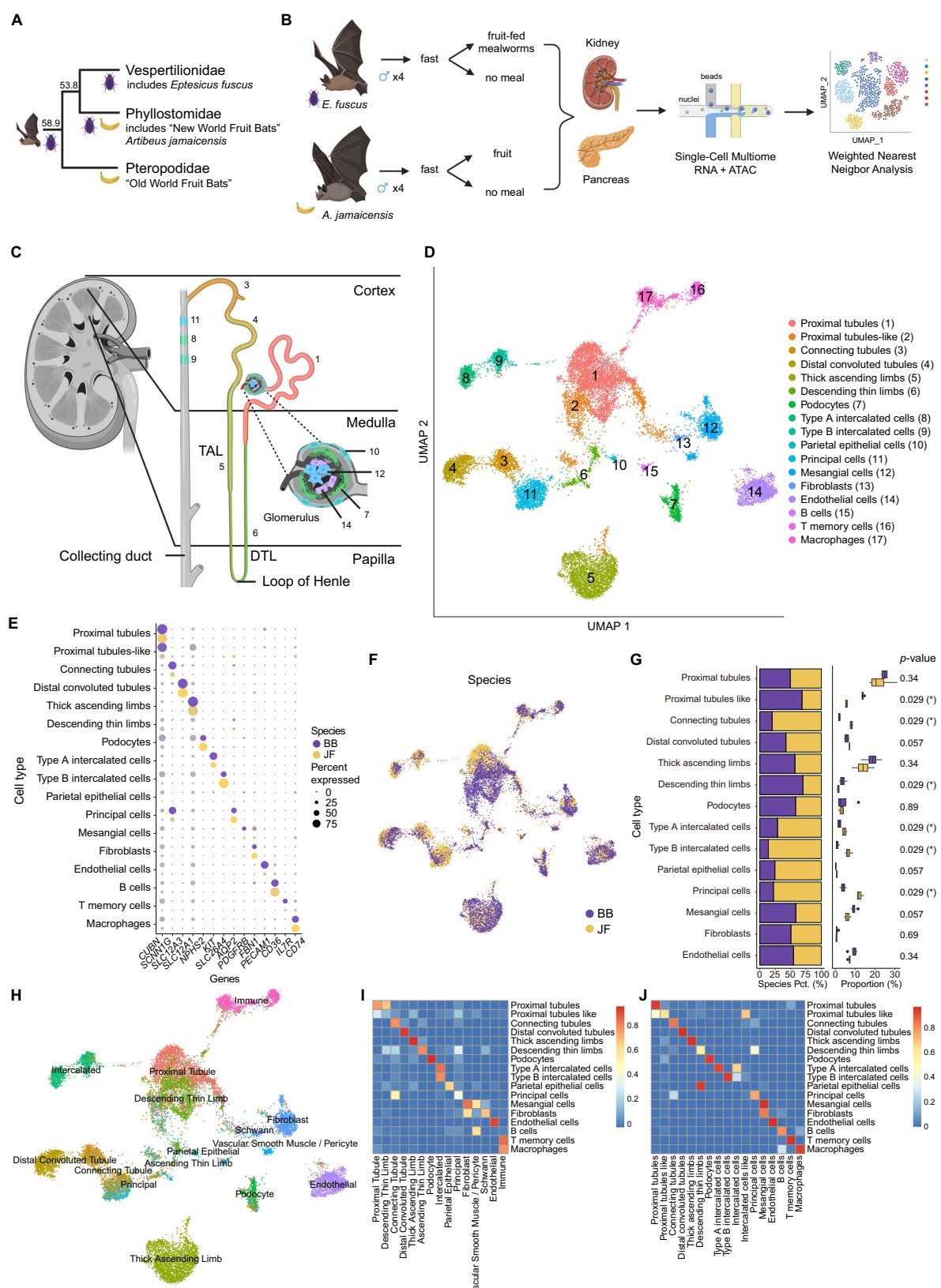

tissue, we used the R package Harmony[48] to correct for batch effects across replicates and applied weighted nearest neighbor analysis[46] (see Methods). To jointly analyze scRNA-seq and scATAC-seq across species for each tissue, we used gene activity scores to maximize the number of shared features for ATAC integration across species. We then employed the same methods used for analyzing within species as

for cross-species analysis (Supplementary Fig. 2, see Methods). Cell-types were identified using canonical mouse and human markers (Supplementary Fig. 3). In addition, we compared our kidney cell-type identifications to insectivorous intermediate horseshoe bats *Rhinolophus affinis*[49]. Because bats diverged from human and mouse lineages approximately 75 million years ago[50], the number of shared markers

**Fig. 1 | Joint scRNA and scATAC profiling of bat kidney. A** Representative phylogenetic tree showing the evolution of frugivory in bats (order: Chiroptera) (created with BioRender.com). The beetle denotes lineage with insectivorous diet, and banana denotes lineage with frugivorous diet. Numbers are in millions of years[41]. **B** Experimental design for joint scRNA and scATAC profiling of bat tissues (see Methods; created with BioRender.com). **C** Diagram of kidney with zoom-in on the nephron. Colors correspond to cell-type colors in (**D**). TAL (Thick ascending limbs), DTL (Descending thin limbs) (created with BioRender.com). **D** Uniform Manifold Approximation and Projection (UMAP) of bat kidney cell-types based on scRNA-seq profiles. **E** Dot plot of marker gene expression across all bat kidney cell-types (Jamaican fruit (JF) and big brown (BB) bat). Color intensity indicates the average expression level across all cells within a cell-type (purple or yellow is high; gray is low). **F** UMAP of bat kidney cell-types by species/dietary phenotype (Jamaican fruit (JF) and big brown (BB) bat). **G** Plot of species percentage across all renal cells (left)

with corresponding two-sided Wilcoxon rank-sum test for differential cell-type abundance between two species ($n = 4$ biologically independent samples for each species) (right) with significant p-values shown in bold font; * <0.05. The boxplots show median and interquartile range (IQR); the lower and upper hinges correspond to the first and third quartiles (the 25th and 75th percentiles). The upper whisker extends from the hinge to the largest value no further than 1.5*IQR from the hinge. The lower whisker extends from the hinge to the smallest value at most 1.5*IQR from the hinge. Pct. = percentage. **H** UMAP of bat kidney cell-types annotated with human adult kidney single-cell reference data from Azimuth[46]. **I** Overlap coefficients of annotations in **h** (horizontal) with our integrated species annotations in **D** (vertical). **J** Overlap coefficients of all annotations found in each species before integration (horizontal) with after integration in (**D**) (vertical). Source data are provided as a Source Data file.

per cell-type identified between bat, human and mouse was limited, restricting our ability to detect novel cell-types that could be unique to bats. We additionally evaluated the homology and functional consistency of cross-species genes that were differentially expressed between bats with inferred bat-human orthologs from TOGA[51] (see Methods), finding that 95.5% and 95.6% of differentially expressed genes in bat kidney and pancreas, respectively, had a predicted ortholog in human (Supplementary Fig. 4a) and 95-96% of differentially expressed genes had a predicted intact or partially intact human orthologous transcript (Supplementary Fig. 4b). We also evaluated pairwise syntenies of each bat and human using MCscan[52] (see Methods) and found that the normalized number of gene pairs for synteny blocks for each bat pairwise synteny was similar to that of mouse and human pairwise synteny (Supplementary Fig. 4c). Of note, while we observed some significant differences in gene expression between fasted and fed states in each species (Supplementary Data 2, 3), these were minimal compared to species gene expression differences (Supplementary Fig. 5a, b) (longer treatments are likely needed for many transcriptional differences between fasted and fed[53]), with the exception of acinar cells, which are known to dominate total mRNA population from the pancreas[54] and permit rapid exocytosis of enzymes for digestion[55]. Given our small sample sizes for treatment (N = 1-2/treatment), we focused our subsequent analyses on species differences (N = 3-4/species). We also did not observe significant differences in cell composition between fasted and fed states in each species by co-varying neighborhood analysis (CNA) (Supplementary Fig. 5c)[56], which was expected as cell-type differences are not likely to transpire following a 30 minute treatment.

## Big brown bat and Jamaican fruit bat kidney composition

For the bat kidney, we initially annotated all major known cell-types using previously reported scRNA-seq markers from human and mouse kidney[57–59]. These included proximal tubules (*CUBN*), connecting tubules (*SCNN1G*), distal convoluted tubules (*SLC12A3*), loop of Henle cells, thick ascending limbs (*SLC12A1*) and descending thin limbs (*SLC14A11*), podocytes (*NPHS2*), type A intercalated cells (*KIT*), type B intercalated cells (*SLC26A4*), principal cells (*AQP2*), mesangial cells (*PDGFRB*), endothelial cells (*PECAM1*), fibroblasts (*FBN1*), and immune cells (*CD36*, *IL7R*, *CD74*) (Fig. 1C–G). Comparison of our cell-type annotations with automated annotations, using single-cell kidney reference databases from humans (Azimuth) (Fig. 1H, I)[46,60] and mouse (Supplementary Fig. 6a, b)[58], showed high similarity. We also annotated a proximal tubules-like cell cluster, as this cluster expresses the proximal tubule markers *MIOX*, *SLC34A1*, and *LRP2* but at lower levels compared to the proximal tubules cluster (Supplementary Fig. 6c). This cluster may be synonymous to injured or regenerative proximal tubule cells in humans[59,61] (Supplementary Data 4). *LRP2, CUBN, SLC12A1, AQP2, PECAM1*, and *CD74* were also similarly used as marker genes for horseshoe bats[49]. To confirm the consistency of cell-type annotations before and after cross-species integration, we compared

the annotations determined separately for each species before integration and the annotations determined jointly after integration and observed high similarity (Fig. 1J).

We detected several cell composition differences between the Jamaican fruit bat and insectivorous big brown bat kidneys. Between the two bats, the big brown bat renal epithelial cell distribution most closely resembled that of the mouse (Supplementary Fig. 6d)[62]. In fruit bats, we found fewer thick ascending limbs (TAL) and significantly fewer descending thin limbs (DTL) cell-types (Fig. 1G), which make up the loop of Henle and are largely found in the renal medulla. The loop of Henle cluster was correlated with the big brown bat by CNA (Supplementary Fig. 6e). Although TAL was not significantly more abundant in the big brown bat by Wilcoxon rank-sum test, likely due to small sample size, a Chi-square test of independence indicates high confidence of TAL enrichment in big brown bats (Supplementary Fig. 6f). This is in line with previous reports that showed that fruit bats have a larger renal cortex and a smaller renal medulla compared to insectivorous bats[16,17,63]. Additionally, we observed Jamacain fruit bats to have significantly more type A intercalated cells, which are involved in acid secretion into the urine, and type B intercalated cells (Fig. 1G, Supplementary Fig. 6f), which mediate bicarbonate secretion while reabsorbing sodium chloride[64], fitting with the high amounts of bicarbonate and the low amounts of sodium in fruit, which has a negative risk of renal acid load[65]. The Jamaican fruit bat also has significantly more principal cells, which reabsorb sodium and excrete potassium, than the big brown bat (Fig. 1G, Supplementary Fig. 6f), in line with fruit containing low sodium and high potassium levels. All of these collecting duct cell-type clusters correlate with the Jamaican fruit bat in CNA (Supplementary Fig. 6e). The fruit bat also has significantly more connecting tubules, which are largely found in the renal cortex, and this cluster correlates with the Jamaican fruit bat by CNA (Fig. 1G, Supplementary Fig. 6e, f). Connecting tubules together with late distal convoluted tubules and the cortical collecting duct (principal cells, type A and B intercalated cells) are often called the aldosterone-sensitive distal nephron (ASDN). Aldosterone increases sodium reabsorption and promotes potassium secretion in the final step of the renin-angiotensin-aldosterone system (RAAS)[66]. As hyperkalemia (excessive potassium in blood) stimulates aldosterone release[66], the greater abundance of connecting tubules in fruit bats is in line with a high potassium diet. The big brown bat has significantly more proximal tubule-like cells as compared to the Jamaican fruit bat, and this cluster also shows a significant association with big brown bats (Fig. 1G, Supplementary Fig. 6e, f). Decreased proximal tubule count is also observed in human diabetic kidneys[67], although proximal tubular growth occurs in early diabetic nephropathy[68]. In summary, we find that Jamaican fruit bat and big brown bat kidneys differ in many nephron components, particularly in proximal tubules-like cells, connecting tubules, the loop of Henle, and the collecting duct.

To further validate these cell composition differences between these species of contrasting diets, we performed

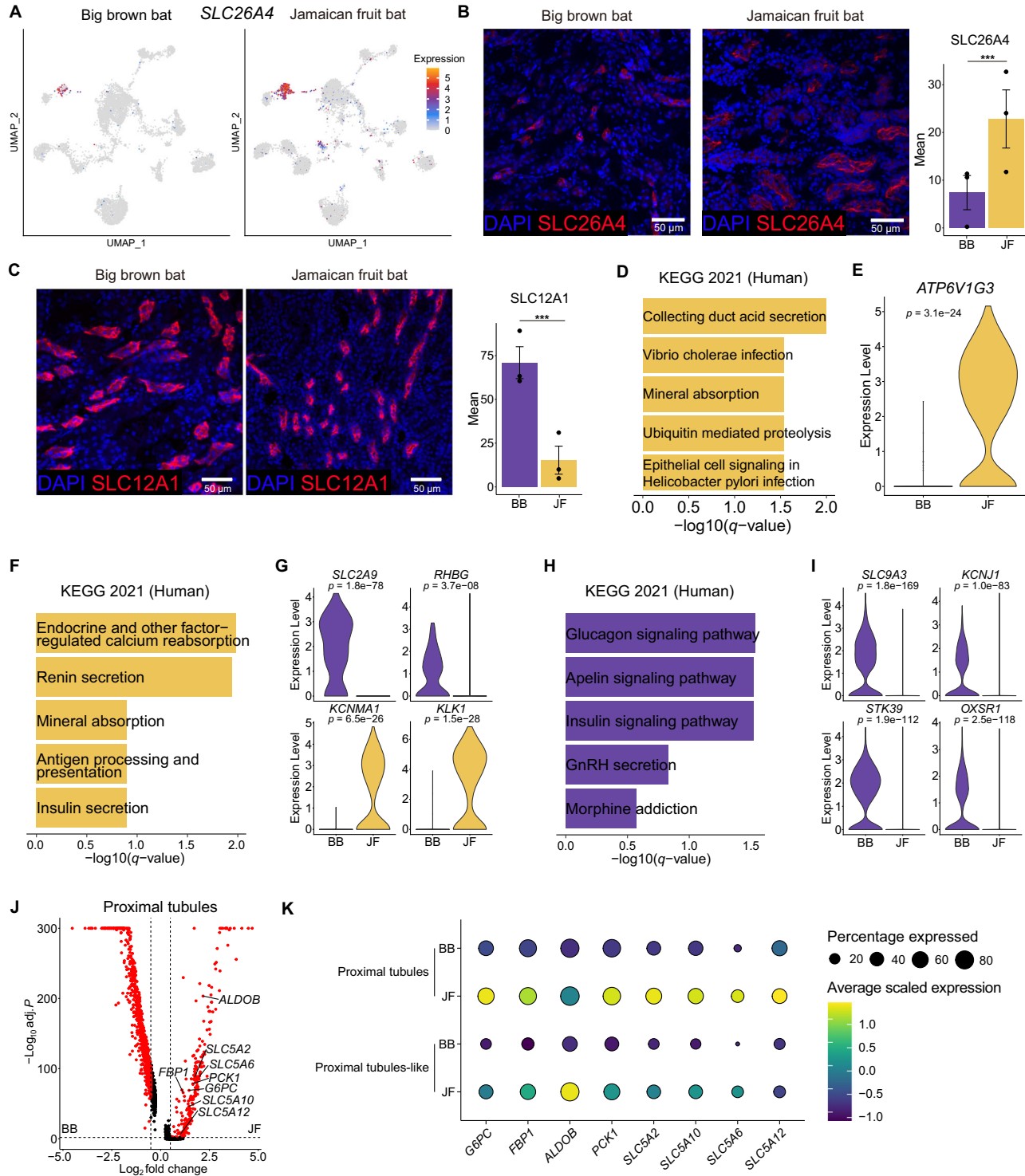

immunofluorescence on various cell-type specific markers on kidney tissue sections from adult big brown bats and Jamaican fruit bats (Supplementary Table 1). For type B intercalated cells, we used an antibody against pendrin, encoded by *SLC26A4*, finding significantly more pendrin-expressing cells in Jamaican fruit bats than in big brown bats (Fig. 2A, B). For TAL, we used an antibody against the Na+/K+/2Cl- co-transporter NKCC2, encoded by *SLC12A1* and found significantly fewer NKCC2-expressing cells in Jamaican fruit bats than in big brown bats (Fig. 2C). For principal cells, we used an antibody against aquaporin 2, encoded by *AQP2*, finding significantly more aquaporin 2-expressing cells in Jamaican fruit bats than in big brown bats (Supplementary Fig. 7a, b). Combined, our

immunofluorescence results validate the single-cell composition differences observed for these cell-types between these species.

## Big brown bat and Jamaican fruit bat kidney gene expression

We next analyzed our scRNA-seq datasets to identify molecular changes between the Jamaican fruit bat and the insectivorous big brown bat that may correlate with their contrasting diets. We conducted gene ontology (GO)[69,70] enrichment analyses for the genes that showed differential expression between the Jamaican fruit bat and the big brown bat kidneys (Supplementary Data 5, 6). For type B intercalated cells, the top KEGG 2021 Human Pathway term for Jamaican fruit bats was collecting duct acid secretion, involving the components

**Fig. 2 | Immunofluorescence and differential gene expression analyses identify traits facilitating frugivory in the Jamaican fruit (JF) bat kidney. A** UMAPs of type B intercalated cell marker gene *SLC26A4* expression in each species (Jamaican fruit (JF) and big brown (BB) bat). **B** (Left) Representative images of *SLC26A4* immunofluorescence (red) in bat kidneys. Nuclei are stained with DAPI (blue). (Right) Quantification of *SLC26A4* immunofluorescence normalized to nuclei in bat kidneys. Results represent arbitrary units of fluorescence (AU) mean ± standard error of the mean (SEM) derived from 3 insectivorous and 3 frugivorous bats ($n = 3$/phenotype, $n = 10$ images/individual [see Methods]). Mixed effects model (two-sided) ****p*-value = 0. Exact *p*-value **C** (Left) Representative images of *SLC12A1* immunofluorescence (red) in bat kidneys. Nuclei are stained with DAPI (blue). (Right) Quantification of *SLC12A1* immunofluorescence normalized to nuclei in bat kidneys. Results represent mean (AU) ± standard error of the mean (SEM) derived from 3 insectivorous and 3 frugivorous bats ($n = 3$/phenotype, $n = 10$ images/individual [see Methods]). Mixed effects model (two-sided) ****p*-value = 0. **D** Bar plots showing Kyoto Encyclopedia of Genes and Genomes (KEGG) Human 2021 pathways enriched in fruit bat type B intercalated cells. *Q*-values were calculated with a one-sided Fisher's exact test and corrected with the Benjamini-Hochberg method. **E** Violin plot of *ATP6V1G3* expression in type B intercalated cells. **F** Bar plots showing KEGG Human 2021 pathways enriched in fruit bat type A intercalated cells. *P*-value calculated with a two-sided Wilcoxon rank-sum test and the Bonferroni correction. **G** Violin plots of *SLC2A9*, *RHBG*, *KCNMA1*, and *KLK1* in type A intercalated cells. *P*-values were calculated with a two-sided Wilcoxon rank-sum test and the Bonferroni correction. **H** Bar plots showing KEGG Human 2021 pathways enriched in insectivore thick ascending limbs. *Q*-values were calculated with a one-sided Fisher's exact test and corrected with the Benjamini-Hochberg method. **I** Violin plots of *SLC9A3*, *OXSR1*, *STK39*, and *KCNJ1* expression in TAL. *P*-values calculated with two-sided Wilcoxon rank-sum test and the Bonferroni correction. **J** Volcano plot showing differentially expressed genes between species in proximal tubules cells. *P*-values calculated with two-sided Wilcoxon rank-sum test and the Bonferroni correction. **K** Dot plots showing the expression of gluconeogenesis and various SLC5 genes in bat proximal tubules and proximal tubules-like cells. Jamaican fruit bat is depicted as JF and big brown bat as BB in the various panels. Source data are provided as a Source Data file.

---

of vacuolar H⁺-ATPase (Fig. 2D). For example, ATPase H⁺ transporting V1 subunit G3 (*ATP6V1G3*) was highly expressed in type B intercalated cells in Jamaican fruit bats and weakly in big brown bats (Fig. 2E). Reabsorption of sodium in type B intercalated cells is fueled by H⁺-ATPase[64], so Jamaican fruit bats most likely require more vacuolar H⁺-ATPase to reabsorb scarce sodium in their diet. For principal cells, the top GO Biological Process terms were all related to maintaining sodium-potassium balance (Supplementary Fig. 7c, d), which includes Jamaican fruit bat upregulated lysine-deficient protein kinases *WNK1* and *WNK4* (Supplementary Fig. 7e). The WNK signaling pathway is regulated by sodium intake via aldosterone as well as insulin[71], and mutations in humans in both these genes lead to hyperkalemia and hypertension[72]. Combined, our single-cell RNA-seq analyses reveal important gene expression differences between these species to high dietary glucose and potassium and low dietary sodium and calcium that occurred within the collecting duct, many of which are associated with causing human disease like hyperkalemia and hypertension.

For type A intercalated cells, the top KEGG 2021 Human Pathway for Jamaican fruit bats were endocrine and other factor-related calcium reabsorption and renin secretion (Fig. 2F). Amongst these genes was kallikrein 1 (*KLK1*), an endocrine-responsive gene that is induced by high glucose[73], that was found to be highly expressed in Jamaican fruit bat type A intercalated cells but not in big brown bats (Fig. 2G). Interestingly, mutations in this gene or its promoter have been found to be associated with hypertension in humans, and its overexpression in mice leads to hypotension, protection from diabetic cardiac damage, renal fibrosis and renal vasodilation, and increased nitrate levels in urine[74], which are found naturally in high concentrations in plants[75]. Furthermore, *KLK1* protects against hyperkalemia after a high potassium load in mice[76]. Another highly expressed gene in the Jamaican fruit bat type A intercalated cells that was not expressed in the big brown bat, was the potassium calcium-activated channel subfamily M alpha 1 (*KCNMA1*) (Fig. 2G). This gene is responsive to renin-mediated regulation of body potassium levels and localizes to the apical membrane of these cells under a high potassium diet to secrete potassium[64]. Furthermore, high potassium stimulates the synthesis and function of the *KCNMA1*-encoded transporter[64], fitting the lack of expression in big brown bats. Increased expression of *KLK1*, *KCNMA1*, and vacuolar H⁺-ATPase in type A intercalated cells in Jamaican fruit bats is likely an adaptation to fruit specialization that helps to address high glucose and high potassium intake.

To determine if TAL gene expression differed substantially between bat species, we used the STRING database[77] to identify genes related to *SLC12A1* and analyzed their expression in our datasets. We found that *STK39* (SPAK), *OXSR1* (OSR1), *KCNJ1* (ROMK), and *SLC9A3* (NHE3), key genes supporting and regulating TAL function in urine concentration[78,79], showed significantly lower expression in Jamaican

fruit bats (Fig. 2H–J). *Stk39* null mice, as well as double knockout of *Stk39* and *Oxsr1*, have increased potassium in the urine, in line with the higher potassium levels observed in fruit bats[80]. Mouse kidney tubule *Oxsr1* knockout, *Kcnj1* knockout, and *Slc9a3* conditional TAL knockout all show lower urine osmolality[81–83], fitting with their observed lower gene expression and the dilute urine found in fruit bats[31]. Loss of function mutations in *SLC12A1* or *KCNJ1* in humans causes Bartter syndrome, which is characterized by hypokalemia, metabolic alkalosis (body pH elevation), polyuria (excess urination), salt wasting, hypercalciuria, and hypotension[84–86]. Combined, our single-cell analyses find that both TAL abundance and gene expression changed substantially in these fruit bats as compared to the insectivorous big brown bat.

We also analyzed our scRNA-seq data for expression of kidney transporters *SLC22A6* (OAT1), *SLC22A12* (URAT1), *SLC2A9* (GLUT9), and *RHBG*, which were reported to be lost in OWFBs[20,21] but intact in NWFBs (Jamaican fruit bat and Honduran yellow-shouldered fruit bat, *Sturnira hondurensis*)[21], and for *MYO6* (myosin VI), which is highly expressed in the OWFB kidney[87]. We found that *SLC22A6*, which is involved in the excretion of organic anions, and *SLC22A12*, which is involved in renal reabsorption of urate, show significantly higher expression in Jamaican fruit bat proximal tubules and proximal tubule-like cells compared to those of the big brown bat (Supplementary Fig. 7e). *SLC22A6* has broad substrate specificity, for both endogenous and exogenous substrates[88], so in NWFBs it could possibly still have a role in the excretion of toxins. Higher expression of *SLC22A12* might be necessary to maintain urate homeostasis in fruit bats, as urate comes from endogenous purines and from diet[89], and fruits are low in purines[90]. *SLC2A9*, which also mediates urate reabsorption in the kidney, was found to be highly expressed in type A intercalated cells in the big brown bat and showed no expression in the Jamaican fruit bat in these cells (Fig. 2G). Lack of *SLC2A9* expression in the collecting duct cells in fruit bats could support urate reabsorption from a high glucose diet, as high glucose concentration inhibits urate reabsorption by this transporter[91]. *RHBG*, which encodes an ammonium transporter, was expressed in type A intercalated cells in big brown bats but largely absent in Jamaican fruit bats (Fig. 2G). Ammonium inhibits sodium reabsorption and potassium secretion[92], so the reduced expression of *RHBG* would accommodate the low sodium-high potassium dietary intake of a frugivore. *MYO6* was upregulated in Jamaican fruit bat podocytes (Supplementary Fig. 7e). Adaptive evolution of *MYO6* in OWFBs was hypothesized to support protein preservation from a protein-scarce diet[87], as mice lacking *MYO6* had reduced endocytosis-mediated protein absorption in proximal tubules and also hypertension[93], which may similarly be supported by upregulation of *MYO6* in NWFB kidney. Together, these results suggest that kidney adaptations are not entirely synonymous across fruit bat lineages.

Bats have been evaluated as a model for diabetes due to their range in lifestyles, diet, and genetic factors[3]. Features of diabetic kidneys include increased gluconeogenesis and glucose reabsorption[94]. We scanned our datasets for gene expression signatures for these processes. We found gluconeogenesis genes *G6PC*, *FBP1*, *ALDOB*, and *PCK1*[94] to all be upregulated in Jamaican fruit bat proximal tubules and proximal tubule-like cells, as well as the glucose reabsorption transporter *SLC5A2* (SGLT2)[95], along with other sodium/glucose cotransport family SLC5 genes *SLC5A10* (SGLT5), *SLC5A6*, and *SLC5A12* (Fig. 2J, K). Increased potassium secretion is also a signature of diabetic nephropathy[67], and we observed many gene expression changes in Jamaican fruit bat kidneys that promote potassium secretion (*KCNMA1*, *STK39*, *OSR1*, *RHBG*) (Fig. 2G, I). Together, our gene expression data indicates that the Jamaican fruit bat kidney shares many characteristics of a diabetic kidney.

## Big brown bat and Jamaican fruit bat kidney gene regulation

To analyze our scATAC-seq datasets and identify gene regulatory differences between the insectivorous big brown bat and Jamaican fruit bat kidneys, we used MACS2[96] to call 209,895 and 188,845 cell-type-specific peaks for these bat kidneys, respectively (Supplementary Data 7, 8). To associate open chromatin regions to genes and human phenotypes, we converted our bat peaks to human genome coordinates (*hg38*)[97], converting 141,779 big brown bat and 124,845 Jamaican fruit bat peaks to *hg38*, respectively. We then used the Genomic Regions Enrichment of Annotations Tool (GREAT)[98] to identify the human target genes associated with these converted bat peaks. The Jamaican fruit bat kidney was found to be highly enriched for human kidney phenotypes, including glucose intolerance, metabolic alkalosis, abnormality of renin-angiotensin system, abnormal circulating renin, and hypokalemia (Supplementary Fig. 7f), fitting with our cell-type and scRNA-seq analyses that showed that the Jamaican fruit bat kidney resembles a diabetic kidney. The big brown bat kidney was only enriched for the human kidney phenotype enuresis nocturna (involuntary urination) (Supplementary Fig. 7f).

Next, to investigate differences in *cis*-regulation between these frugivorous and insectivorous bats, we performed differential motif enrichment analyses on renal epithelial cells using Analysis of Motif Enrichment (AME)[99]. For collecting duct cells, we found two clusters of differentially enriched motifs, which separated Jamaican fruit bats and big brown bats (Fig. 3A). The Jamaican fruit bat collecting duct cells were enriched for *NKX*, *FOX*, and *NFAT* TFBS (Fig. 3A). *NKX* and *FOX* TFs are broadly expressed and important for development. *NFATc* and *NFAT5* are regulated by tonicity and protect the kidney from osmotic stress by activating genes that contribute to cellular accumulation of organic osmolytes, such as *AQP2*[100,101]. The high water intake frugivores get from fruit may contribute to higher osmotic stress in the kidney, and *AQP2* was upregulated in frugivore principal cells (Supplementary Fig. 7b). The big brown bat collecting duct cells were enriched for *E2F* and *EGR* motifs (Fig. 3A). *E2F* TFs are involved in cell cycle regulation[102]. *EGR1* knockdown may alleviate renal injury in diabetic kidney disease (DKD) by downregulating renin and RAAS[103]. The big brown bat collecting duct cells were also enriched for *KLF15* (Fig. 3A), which protects against podocyte injury[104], and *TFAP2A*, which maintains adult mouse medullary collecting duct structure[105]. Consistent with the medullary composition differences between these bat species, big brown bats have upregulated *TFAP2A* expression in principal cells (Supplementary Fig. 7g). Collectively, differential motif enrichment analyses on bat collecting duct cells suggest the importance of osmoregulation in fruit bats and of medulla maintenance in insectivorous bats.

We also separated proximal tubules, proximal tubule-like cells, distal convoluted tubules, TAL and DTL for differential motif enrichment and found multiple TFBS clusters that differentiate Jamaican fruit bat and big brown bats, including many of the same TFs identified in collecting duct cells (Fig. 3A). High glucose-induced inactivation of

*FOXO1* likely contributes to DKD pathogenesis[106], and *FOXO1* was downregulated in Jamaican fruit bat distal convoluted tubules, connecting tubules, TAL and DTL (Supplementary Fig. 7g). Enriched in fruit bat tubules, *FOXO3* protects against kidney injury in T2D nephropathy[107], and the *FOXO3* motif was also enriched in human diabetic proximal tubules[94]. Jamaican fruit bat tubules also shared enrichment with collecting duct cells for vitamin D receptor (*VDR*) motifs (Fig. 3A). *VDR* is highly expressed in renal tubules and plays a renoprotective role in a diabetic mouse model[108]. *SOX* TFs were more enriched in Jamaican fruit bat tubules than in collecting duct cells (Fig. 3A). *SOX6* is upregulated after a low sodium diet in mouse and controls renin secretion[109]. In addition to *TFAP2A* motifs, big brown bat tubules were enriched for *TFAP2B* motifs (Fig. 3A). *TFAP2B* is critical for the formation and function of distal convoluted tubules[105] and was upregulated in big brown bat distal convoluted tubules (Supplementary Fig. 7g). Big brown bat tubules were notably enriched for *POU* TFBS, some of which were shared with Jamaican fruit bat TAL (Fig. 3A). *POU* TFs have multiple roles in development and neuroendocrine function[110]. In summary, kidney TFBS differential enrichment between these insectivorous and frugivorous bats identified key TFs involved in diet, with the Jamaican fruit bats demonstrating some diabetic-associated motif signatures.

To further understand gene regulatory differences between these frugivorous and insectivorous bat kidneys, we conducted multi-omics gene regulatory network (GRN) analysis on renal epithelial cells with Pando[111]. To infer global GRNs, Pando simultaneously utilizes co-expression of TF genes and target genes and co-accessibility of TFBSs calculated from gene expression and chromatin accessibility profiles. Jamaican fruit bat kidneys had GRNs enriched for acid-base and electrolyte homeostasis and glucose pathways, while big brown bat kidneys had GRNs enriched for transcriptional and signal transduction pathways (Supplementary Fig. 8a–c). For example, *SLC13A2*, which encodes the sodium-dependent dicarboxylate transporter 1 (NaDC1), is a critical regulator of acid-base homeostasis and blood pressure and displays high centrality in the Jamaican fruit bat kidney, but not in the big brown bat kidney (Supplementary Fig. 8b). *ADGRF5*, a regulator of acid-base homeostasis, displays high centrality in the big brown bat kidney (Supplementary Fig. 8b). Adhesion-GPCR Gpr116 (*ADGRF5*) inhibits renal acid secretion by regulating vacuolar $H^+$-ATPase expression[112], which may support the presence of collecting duct acid secretion found as an enriched pathway in Jamaican fruit bats (Fig. 2D) but not in big brown bats. We also observed notable differences within the networks of genes that have high strength centrality in both Jamaican fruit bats and big brown bats. In the Jamaican fruit bat network of *NR3C2*, which encodes the mineralocorticoid receptor responsible for aldosterone-regulation of electrolyte and blood pressure balance[113], we found potassium transporter *KCNIP1* and vacuolar $H^+$-ATPase subunits *ATP6V0A4* and *ATP6V1B1* to be upregulated and sulfate transporter *SLC13A1* to be downregulated (Supplementary Fig. 8d). We also observed insulin receptor substrate-1 (*IRS1*), *TBC1D4*, which regulates insulin-stimulated glucose uptake by controlling GLUT4 translocation[114], and *TOX*, a transcriptional regulator in T1D[115] and associated with diabetic nephropathy[116], to be upregulated in our fruit bat. The major gluconeogenesis regulator *PCK1* and ketohexokinase gene *KHK*, which encodes the first enzyme to catabolize dietary fructose[117], were downregulated. In contrast, the *NR3C2* network in big brown bat kidneys does not have these sugar regulation genes and instead involves different solute transporters than found in the Jamaican fruit bat network (Supplementary Fig. 8d). Together, these results indicate that the GRNs of Jamaican fruit bats and big brown bats strongly differ in the kidney.

We next sought to utilize our integrative single-cell datasets to survey diabetes-associated regions in bat kidneys. Examination of *KLK1*, which regulates blood pressure, showed a substantial increase in chromatin accessibility in the collecting duct cells as well as proximal

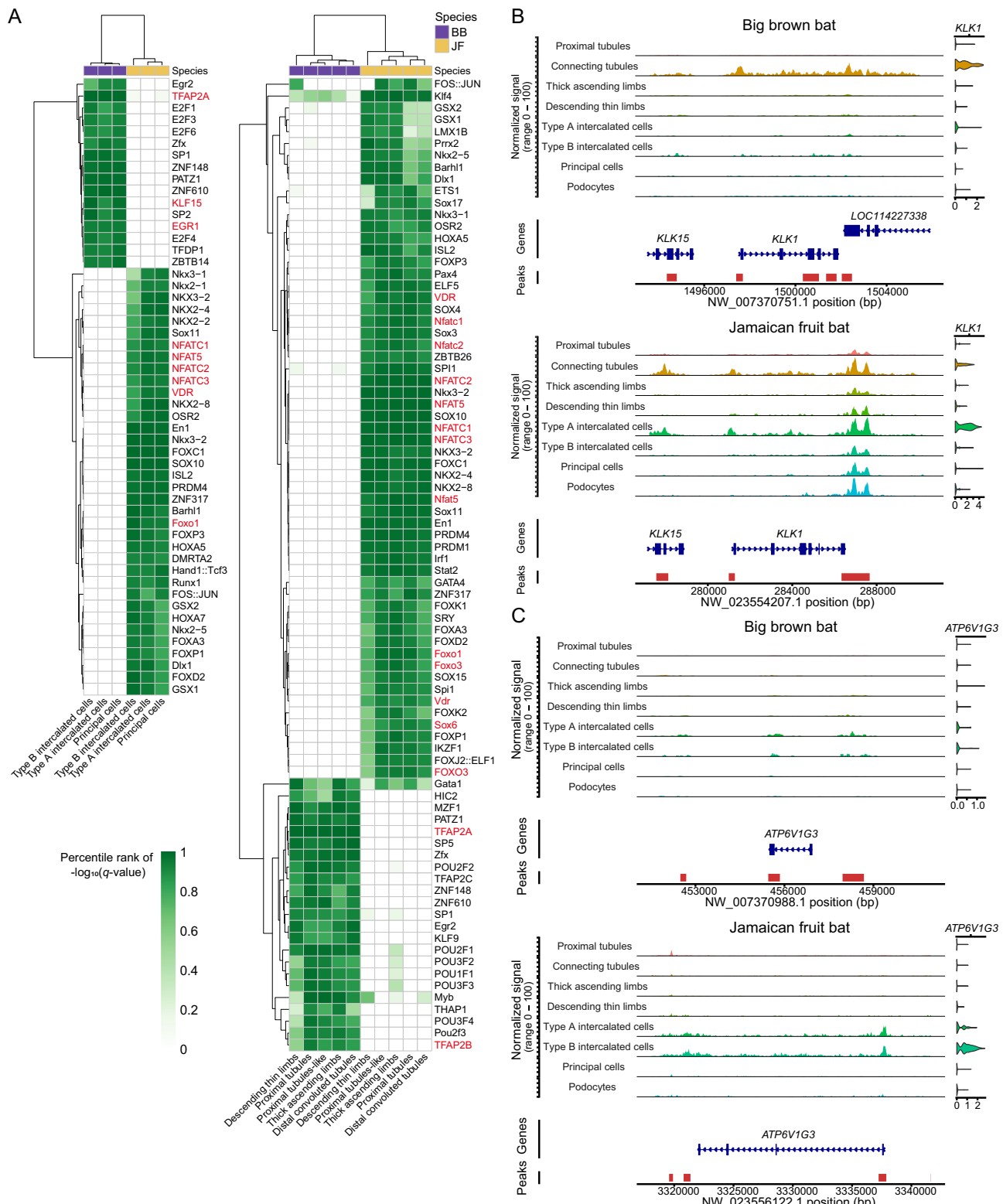

**Fig. 3 | scATAC analyses reveal TFBS and chromatin accessibility in bat kidneys.** **A** (Left) Heatmap of TF motifs enriched in bat kidney collecting duct cells. (Right) Heatmap of TF motifs enriched in bat renal tubule and limb cells. *P*-values calculated with one-sided Fisher's exact test and corrected with Bonferroni correction.

**B**, **C** scATAC-seq coverage plots of *KLK1* (**B**) and *ATP6V1G3* (**C**) in bats. SCTransform-normalized expression plot visualized on the right by cell-type. Jamaican fruit bat is depicted as JF and big brown bat as BB in the various panels.

tubules, TAL, and podocytes in the Jamaican fruit bat (Fig. 3B). *ATP6V1G3*, which is a biomarker for diabetic nephropathy[118], showed greater chromatin accessibility, particularly in the promoter, in Jamaican fruit bat type A and type B intercalated cells (Fig. 3C). The big

brown bat *ATP6V1G3* promoter is predicted to bind *RREB1*, which represses RAAS through the angiotensin gene and is associated with type 2 diabetes (T2D) end-stage kidney disease[119]. Angiotensin is known to stimulate vacuolar H⁺-ATPase activity in intercalated cells[120],

so the lack of the *RREB1* motif in the Jamaican fruit bat *ATP6V1G3* promoter may allow greater activation of RAAS effects. In addition, *RREB1* expression is lower in Jamaican fruit bat proximal tubules and proximal tubules-like cells (Supplementary Fig. 7g). We also observed similar chromatin accessibility trends between human diabetic proximal tubules and controls[94] in bats (Supplementary Fig. 9). For example, *PCK1* demonstrated decreased accessibility near the end of the gene body in diabetic proximal tubules[94], which was also observed in the Jamaican fruit bat kidney (Supplementary Fig. 9a). The Jamaican fruit bat sequence homologous to the big brown bat peak following the end of the *PCK1* gene body is predicted to bind transcriptional repressor *ZNF331*[121], fitting with the decreased accessibility of this region in Jamaican fruit bat proximal tubules. Together, our analyses indicate that Jamaican fruit bat kidneys display many features similar to human diabetic kidneys.

### Big brown bat and Jamaican fruit bat pancreas composition

We next set out to analyze our multiome data for the bat pancreas. Similar to the kidney, we initially annotated all major cell-types using markers from previously reported scRNA-seq datasets from human and mouse pancreases[57,122,123]. We detected all major known cell-types found in human and mouse pancreas: acinar cells (*CPA2*), ductal cells (*SLC4A4*), beta cells (*INS*), alpha cells (*GCG*), delta cells (*SST*), endothelial cells (*SLCO2A1*), active pancreatic stellate cells (aPSCs; *LAMA2*), quiescent pancreatic stellate cells (qPSCs; *COL3A1*), and immune cells (*ACTB*, *CD74*, *IL7R*, *MRC1*) (Fig. 4A–E). We identified gamma cells and epsilon cells by other known markers (*CHGA*[124]; and *ACSL1*[123], respectively), as many major markers were not shared across species. The overwhelming majority of immune cells were found in only one Jamaican fruit bat (Supplementary Fig. 10a, b). Due to this immune cell bias and our low sample size, we focused only on non-immune cell types in our subsequent analyses. We observed a high similarity between our cell-type annotations and automated annotations using human single-cell pancreas reference data (Fig. 4F, G)[123]. Consistent with the adult human pancreas, acinar cells showed clear heterogeneity in the bat pancreas, with secretory acinar cells (acinar-s) expressing higher levels of digestive enzyme genes (*CPA1*, *PLA2G1B*, *SYCN*, *KLK1*, *CLPS*) and idling acinar cells (acinar-i) expressing lower levels of digestive enzyme genes (Supplementary Fig. 10c). We compared the annotations determined separately for each species before integration and the annotations determined jointly after integration and observed high similarity (Fig. 4H).

Fruit bats are reported to have an expansion of endocrine tissue in the pancreas[18,19]. Indeed, of the pancreatic cell-types, we found that the Jamaican fruit bat pancreas is about 26% endocrine, whereas the big brown bat pancreas is about 12% (Supplementary Fig. 10d). Both beta and alpha cell clusters are largely correlated with the Jamaican fruit bat by CNA (Supplementary Fig. 10e). As the smaller Jamaican fruit bat sample size (N = 3) precluded cell composition significance between species with Wilcoxon rank-sum test, we also conducted Chi-square tests of independence on Pearson residuals. We found, with high confidence, that both beta and alpha cells are more abundant in the Jamaican fruit bat (Supplementary Fig. 10f). Beta and alpha cells regulate blood glucose levels via insulin and glucagon, respectively, and their increased numbers are in line with the need to respond to a high glucose diet and tight blood glucose regulation in fruit bats, which have been shown to robustly regulate blood sugar in intraperitoneal glucose tolerance tests[3,22–24]. We found acinar cells to be more abundant in the big brown bat (Supplementary Fig. 10e), with Jamaican fruit bats having fewer acinar cells than big brown bats with high confidence (Supplementary Fig. 10f). Acinar cells, which produce digestive enzymes for storage and secretion[125], comprise about 75% of big brown bat pancreatic cells, which is largely consistent with that of the human adult pancreas[123], compared to 52% in Jamaican fruit bats (Supplementary Fig. 10d). The substantial reduction in acinar cells in Jamaican

fruit bats could accommodate the increase of endocrine cells. The Jamaican fruit bat also has more ductal cells with high confidence (Supplementary Fig. 10f), which secrete enzymes from acinar cells into the duodenum and secrete bicarbonate to neutralize stomach acidity[126], although the overall percentage of ductal cells in the Jamaican fruit bat pancreas (11%) corresponds with that of humans (10%)[126]. In summary, our results suggest that compared to the insectivorous big brown bat, the expanded endocrine pancreas in the Jamaican fruit bat is largely attributable to increased numbers of beta and alpha cells, with proportionally fewer exocrine acinar cell numbers that likely compensate for the endocrine expansion.

To validate our single-cell composition results, we performed immunofluorescence on pancreas tissue sections from the same bat species (Supplementary Table 1). For beta cells, we used an antibody against insulin, encoded by *INS*, and found that there were significantly more insulin-expressing cells in Jamaican fruit bats than in big brown bats (Fig. 5A). For alpha cells, we used an antibody against glucagon, encoded by *GCG*, and found that there were also significantly more glucagon-expressing cells in Jamaican fruit bats than in big brown bats (Fig. 5B). Our immunofluorescence results thus validated our observed single-cell composition differences for endocrine cells.

### Big brown bat and Jamaican fruit bat pancreas gene expression

To identify pancreatic gene expression differences between these two bat species that could be associated with frugivory, we performed gene enrichment analysis on all cell-types using differentially expressed genes between the Jamaican fruit bat and the big brown bat (Supplementary Data 9, 10). As acinar-i cells appeared largely separated by species according to CNA (Fig. 4E, Supplementary Fig. 10e), we did sub-clustering to analyze phenotypic differences within this cell-type (Supplementary Fig. 11a), and found that cluster 0 (C0) was predominantly big brown bat-based and cluster 1 (C1) to be predominantly Jamaican fruit bat-based. C0 was enriched for KEGG 2021 Human Pathway terms related to protein synthesis and secretion, fitting with the increased protein composition in insects. C1 was enriched for diabetic cardiomyopathy as well as oxidative phosphorylation (Supplementary Fig. 11b, c), which is notably shared with human T2D acinar cells[127]. Ductal cells, which produce alkaline-high pancreatic secretions and are regulated by calcium signaling[128], were enriched in Jamaican fruit bats for calcium signaling pathway genes, including *PDE1A*, *EGFR*, and *ERBB4* (Supplementary Fig. 11d–f). Fruit bats may stimulate more pancreatic secretions due to their low calcium diet to quickly digest carbohydrates and rapidly fuel their metabolism[25].

We next determined if genes suggested to be adaptive for frugivory in OWFBs were differentially expressed between our NWFBs and insectivorous big brown bats. *FAM3B* and *FFAR3*, metabolic genes involved in insulin metabolism and signaling, were reported to be lost in OWFB genomes[20,21]. We observed both genes to be expressed in the NWFB pancreas (Supplementary Fig. 11f). Expression of *FAM3B* was sparse throughout the NWFB pancreas as well as the big brown bat pancreas. *FFAR3*, an inhibitor of insulin secretion[129,130], was expressed in big brown bat beta cells and was nearly absent in Jamaican fruit bats (Supplementary Fig. 11f). Weak expression of *FFAR3* in NWFB pancreas is consistent with the hypothesis that loss of this gene in OWFBs is adaptive, as fruit bats secrete large amounts of insulin[19].

We then analyzed the expression of glucose transporters (GLUTs) in pancreatic cells. We first examined the main glucose transporter in beta cells[131], *SLC2A2* (GLUT2), which was previously shown to have a fruit bat-specific 11 bp deletion in its proximal promoter[3], and found that it was not differentially expressed between our insectivorous and frugivorous bats (Supplementary Fig. 11f). Other GLUTs identified in human pancreatic cells also did not show differential expression between big brown bats and Jamaican fruit bats, with the exception of *SLC2A13* (GLUT13), a H+/myo-inositol transporter found in both human endocrine and exocrine cells[131]. *SLC2A13* showed significantly

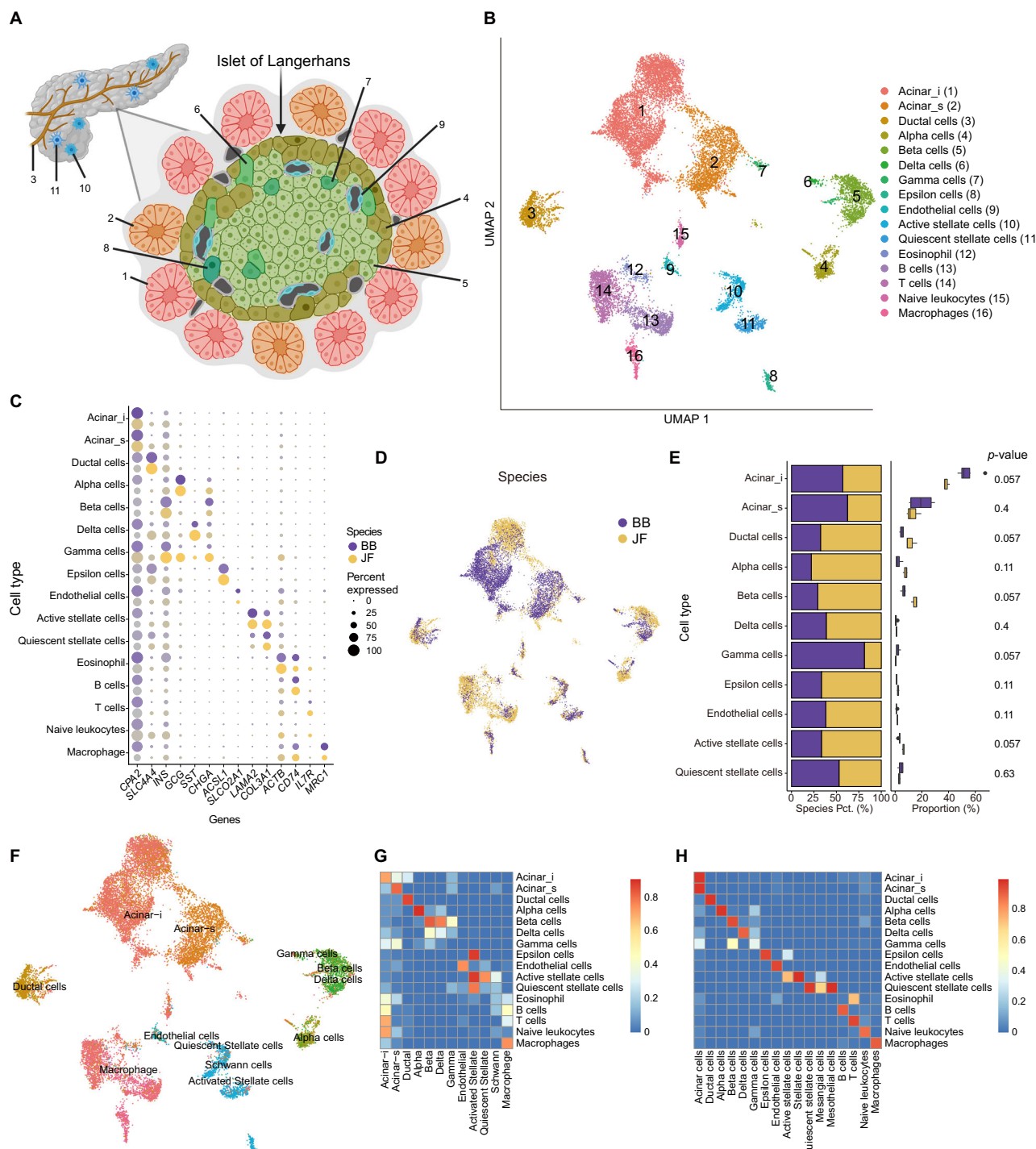

**Fig. 4 | Joint scRNA and scATAC profiling of the bat pancreas. A** Diagram of pancreas with zoom-in on the Islet of Langerhans. Colors correspond to cell-type colors in **b** (created with BioRender.com). **B** UMAP of bat pancreas cell-types based on scRNA-seq data. **C** Dot plot of marker gene expression across all bat pancreas cell-types. Color intensity indicates the average expression level across all cells within a cell-type (purple or yellow is high, gray is low). **D** UMAP of bat pancreas cell-types by species/dietary phenotype. **E** Plot of species percentage across all pancreatic cells (left) with corresponding two-sided Wilcoxon rank-sum test for differential cell-type abundance between two species ($n = 3$ biologically independent samples for the frugivore bats and $n = 4$ biologically independent samples for the insectivore bats). The boxplots show median and interquartile range (IQR); the

lower and upper hinges correspond to the first and third quartiles (the 25th and 75th percentiles). The upper whisker extends from the hinge to the largest value no further than 1.5*IQR from the hinge. The lower whisker extends from the hinge to the smallest value at most 1.5*IQR from the hinge. (right). Pct. percentage. **F** UMAP of bat pancreas cell-types automatically annotated with human pancreas single-cell reference data[123]. **G** Overlap coefficients of auto-annotations in (**G**) (horizontal) with our integrated species annotations in (**B**) (vertical). **H** Overlap coefficients of all annotations found in each species before integration (horizontal) with after integration in (**B**) (vertical). Jamaican fruit bat is depicted as JF and big brown bat as BB in the various panels. Source data are provided as a Source Data file.

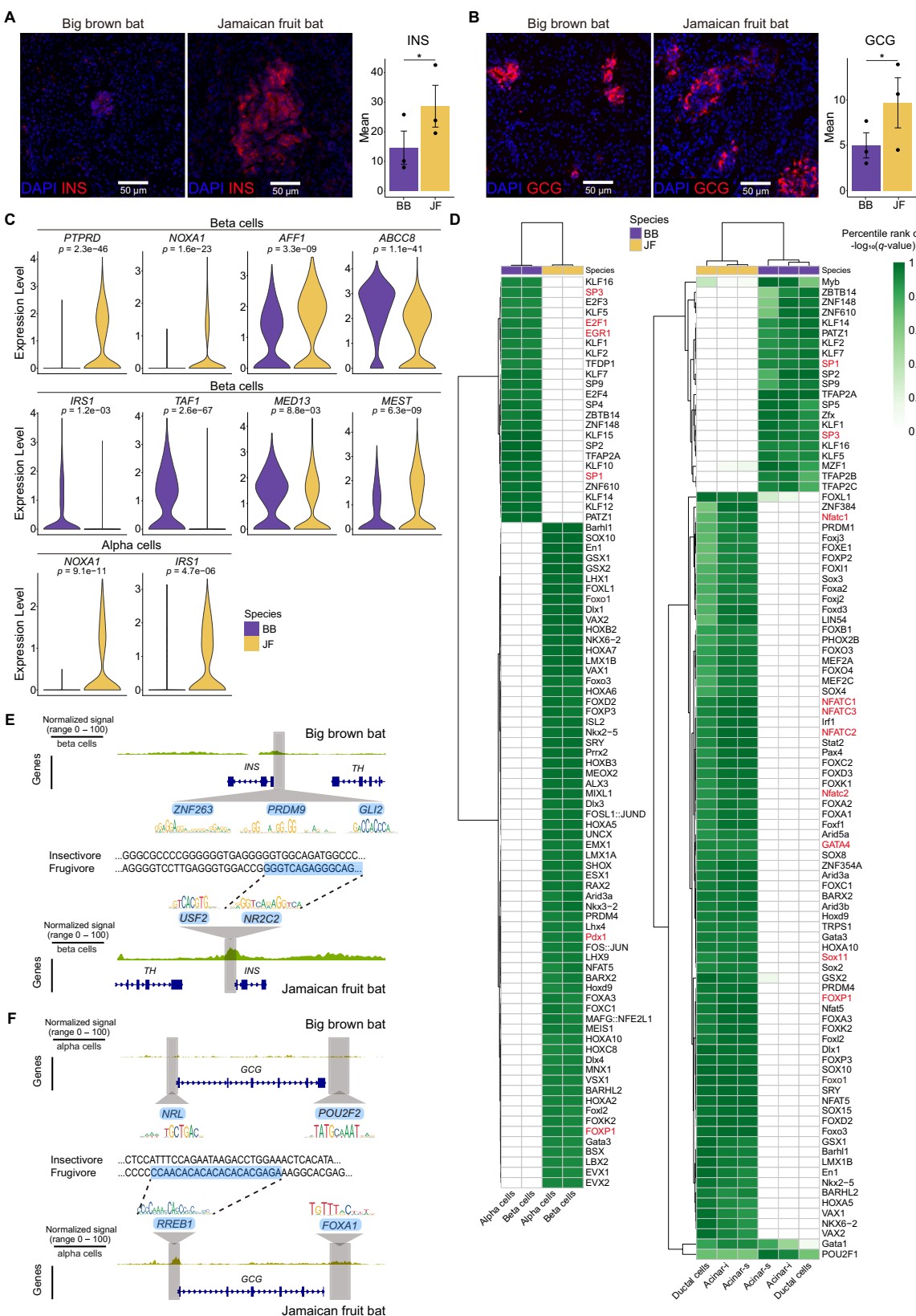

higher expression in Jamaican fruit bat exocrine acinar-i and ductal cells (Supplementary Fig. 11f). Myo-inositol is found in fruits like cantaloupe, which was consumed by our fruit bats, and is produced in tissues such as the kidneys[132]. Increased sugar intake and diabetes increase the need for myo-inositol[132]. Increased expression of *SLC2A13* in Jamaican fruit bat exocrine pancreas complements the upregulation

of myo-inositol oxygenase enzyme (*MIOX*), which catalyzes myo-inositol breakdown, in the liver of the same fruit bat species we analyzed (Jamaican fruit bat) but not in four insectivorous bat species[133]. In line with these findings, we also observed upregulation of the myo-inositol transporter *SLC5A10*[134] in the Jamaican fruit bat kidney proximal tubules (Fig. 2K).

**Fig. 5 | scRNA and scATAC analyses depict both exocrine and endocrine dietary differences between the big brown bat (BB) and Jamaican fruit (JF) bat pancreases. A** (Left) Representative images of *INS* immunofluorescence (red) in bat pancreases. Nuclei are stained with DAPI (blue). (Right) Quantification of *INS* immunofluorescence normalized to nuclei in bat pancreases. Results represent arbitrary units of fluorescence (AU) mean ± standard error of the mean (SEM) derived from 3 insectivorous and 3 frugivorous bats (*n* = 3/phenotype, *n* = 10 images/individual [see Methods]). Mixed effects model (two-sided) *\*p*-value = .002. **B** (Left) Representative images of *GCG* immunofluorescence (red) in bat pancreases. Nuclei are stained with DAPI (blue). (Right) Quantification of *GCG* immunofluorescence normalized to nuclei in bat pancreases. Results represent arbitrary units of fluorescence (AU) mean ± standard error of the mean (SEM) derived from 3 insectivorous big brown bats and 3 Jamaican fruit bats (*n* = 3/phenotype, *n* = 10

images/individual [see Methods]). Mixed effects model (two-sided) *\*p*-value = .008. **C** Violin plots of diabetes-associated genes in bat endocrine cells. *P*-values calculated with two-sided Wilcoxon rank-sum test and the Bonferroni correction. **D** (Left) Heatmap of TF motifs enriched in bat pancreatic endocrine cells. (Right) Heatmap of TF motifs enriched in bat pancreatic exocrine cells. *P*-values calculated with one-sided Fisher's exact test and corrected with Bonferroni correction **E** scATAC-seq coverage plots of *INS* in bat pancreases with predicted TFBS highlighted in gray. Aligned sequence zoom-in on *NR2C2* motif with differing nucleotides depicted in bold. **F** scATAC-seq coverage plots of *GCG* in bat pancreases with predicted TFBS highlighted in gray. Aligned sequence zoom-in on *RREB1* motif with differing nucleotides depicted in bold. Jamaican fruit bat is depicted as JF and big brown bat as BB in the various panels. Source data are provided as a Source Data file.

---

Features of a diabetic pancreas include a reduced exocrine pancreas and loss of beta cells, with differences in the presence of insulitis and islet amyloidosis between T1D and T2D[135]. We compared our gene expression data to reported single-cell datasets from the diabetic human pancreas[127,136,137]. Differentially expressed genes between human T2D beta cells and normal beta cells, determined from three separate single-cell studies with a combined 19 T2D donors and 33 non-diabetic donors[138], included *TAF1*, *PTPRD*, *NOXA1*, *MED13*, *AFF1*, and *MEST*. These genes were also differentially expressed in our datasets between insectivorous big brown bat and Jamaican fruit bat beta cells (Fig. 5C). *PTPRD* is involved in insulin signaling and variants within this gene were found to be associated with gestational diabetes risk[139,140]. This gene was found to be upregulated in T2D beta cells and in our fruit bat beta, alpha, and ductal cells. *AFF1*, which has circular RNAs that control beta cell apoptosis[141], was also upregulated in both T2D and Jamaican fruit bat beta and acinar cells. Chronic hyperglycemia is known to upregulate NAPDH oxidase (NOX) genes[142], and *NOXA1* (NOX activator 1) was found to be upregulated in our fruit bat beta, alpha, and acinar-i cells. We also observed the monogenic diabetes risk gene *ABCC8*, a regulator of potassium channels and insulin release, whose loss of function causes hyperinsulinism (excessive insulin secretion) and polyuria[143], to be downregulated in fruit bat beta cells. In addition, *IRS1*, deletion of which in mice causes hyperinsulinism but not diabetes[144], was found to be downregulated in Jamaican fruit bat beta cells and upregulated in Jamaican fruit bat alpha and acinar-i cells. Taken together, the Jamaican fruit bat pancreas exhibits gene expression changes that are known to elevate and tightly regulate insulin secretion and signaling in response to a high sugar diet, many of which correspond to human diabetic gene expression dysregulation.

## Big brown bat and Jamaican fruit bat pancreas gene regulation

To analyze open chromatin regions specific to cell-types in the pancreas, we called peaks with MACS2[96], finding 187,421 and 281,823 cell-type-specific peaks in insectivorous big brown and Jamaican fruit bat pancreases, respectively (Supplementary Data 11, 12). To associate open chromatin regions to genes and human and mouse phenotypes, we identified 111,619 and 184,360 peaks from the big brown bat and Jamaican fruit bat genomes, respectively, in humans (*hg38*). Analysis of these human peaks using GREAT[98] found the big brown bat pancreas to be enriched for many mouse phenotype terms involving fat metabolism, including abnormal lipolysis, impaired lipolysis, increased inguinal fat pad weight, increased white fat cell number, and abnormal white fat cell number (Supplementary Fig. 11g), which could be indicative of adaptation to a higher fat diet from insects. The Jamaican fruit bat pancreas was most enriched for human pancreas phenotype metabolic alkalosis (Supplementary Fig. 11g), indicative of the high bicarbonate diet of Jamaican fruit bats.

We then used AME[99] to identify differentially enriched TFBS in scATAC-seq peaks between big brown bats and Jamaican fruit bats in endocrine and exocrine cell-types. We found that there are two clusters of TFBSs that show differential enrichment across alpha and beta

cells of the two bats (Fig. 5D). Similar to renal epithelial cells, *FOX* motifs, such as *FOXP1*, were enriched in Jamaican fruit bat endocrine cells (Fig. 5D). *FOXP1* is required for alpha cell proliferation and function[145], and *FOXP1* is upregulated in Jamaican fruit bat alpha cells as well as beta cells (Supplementary Fig. 11f). Jamaican fruit bat endocrine cells were also strongly enriched for *HOX* gene TFBS (Fig. 5D), which play roles in pancreatic development[146]. TFBS for *PDX1*, which regulates endocrine and exocrine cell development and binds the mouse insulin enhancer[147], were notably enriched in Jamaican fruit bat endocrine cells (Fig. 5D). Big brown bat endocrine cells were enriched for *SP* and *KLF* motifs (Fig. 5D), TFs that regulate the pathophysiology of the digestive system, many of which maintain beta cell function and control oxidative stress response genes[148]. Similar to the kidney, big brown bat endocrine cells were also enriched for *E2F* motifs, including *E2F1* (Fig. 5D), which regulates glucagon-like peptide-1 during insulin secretion[149], and *EGR1*, which regulates glucose homeostasis and pancreatic islet size[150]. Together, TFBS analyses demonstrate that the TF families utilized in endocrine pancreas regulation strongly differ between these bats.

Two clusters of TFBSs were also annotated for exocrine cell-types, indicating again that there is a divergence in TFBS usage between these two bat species (Fig. 5D). In addition to endocrine cells, Jamaican fruit bat exocrine cells were enriched for *FOX* TFBS (Fig. 5D). Jamaican fruit bat exocrine cells were also enriched for *NFAT* motifs, which were enriched in the Jamaican fruit bat kidney (Fig. 3A). *NFATc*1-3 are expressed in acinar cells and promote pancreatic growth[151]. *GATA4*, that is expressed in acinar cells and controls mouse pancreas organogenesis, was also enriched in Jamaican fruit bat exocrine cells (Fig. 5D). Similar to Jamaican fruit bat renal tubules, *SOX* motifs, which are necessary for developmental and physiological regulation of the pancreas[152], were highly enriched in Jamaican fruit bat exocrine cells (Figs. 3A, 5D), including *SOX11*, which *SOX11*-deficient mouse embryos exhibit pancreatic hypoplasia[152]. In addition to endocrine cells, big brown bat exocrine cells were enriched for *SP* and *KLF* TFBS (Fig. 5D). *SP1* maintains acinar cell function and inhibits proliferation, and both *SP1* and *SP3* regulate the expression of the secretin receptor gene[148]. Big brown bat exocrine cells were also enriched for *TFAP* TFBS (Fig. 5D), which were also identified in the kidney (Fig. 3A). *TFAP2B* was identified as an obesity-associated loci[153]. Taken together, these bat species differ strongly in TF usage in the pancreas.

We next conducted multi-omics GRN analysis on endocrine and exocrine cell-types with Pando[111]. We found Jamaican fruit bat and big brown bat pancreases to both have GRNs enriched for transcriptional and neuronal pathways (Supplementary Fig. 12a–c). Potassium channel gene *KCNB2* plays a role in regulating glucose-stimulated insulin secretion[154] and displays high centrality in Jamaican fruit bats, while potassium channel gene *KCNJ6*, which may also contribute to regulation of insulin secretion, displays high centrality in big brown bats (Supplementary Fig. 12b). Upregulated in Jamaican fruit bat cells, *PTPRD*, which is involved in insulin signaling and associated with gestational diabetes risk[139,140], also displayed high centrality in the

Jamaican fruit bat (Fig. 5C; Supplementary Fig. 12b). Similar to our previous GRN kidney analysis, we observed differences within the networks of genes that have high strength centrality in both Jamaican fruit bats and big brown bats. In the Jamaican fruit bat network of *ABCC8*, which was downregulated in Jamaican fruit bat beta cells (Fig. 5C), *ABCC8* was upregulated by *NEUROD1*, which both regulates endocrine cell maturation and function[155,156], and was downregulated by *GATA4*, which is required for pancreatic development and increases glucose-dependent insulinotropic polypeptide expression in a mouse beta cell line[157,158] (Supplementary Fig. 12d). *ABCC8* was also downregulated by *GATA4* cofactor and insulin signaling regulator *ZFPM2*[159,160]. The big brown bat network of *ABCC8* involves fewer and different genes but shares *ABCC8* upregulation by *DACH1* and *RFX3*, both of which play roles in islet cell development and *RFX3* additionally regulates beta cell function and glucokinase expression[161,162] (Supplementary Fig. 12d). Together, these results indicate that the Jamaican fruit bats and big brown bats share GRNs but the genes involved differ.

Next, we examined specific diabetes-associated regions for gene regulatory differences between the insectivorous big brown bats and Jamaican fruit bats. We found more open chromatin in the promoter of *INS*, which was more highly expressed in Jamaican fruit bat beta cells (Fig. 5E, Supplementary Fig. 11f, h). In the promoter of *GCG*, as well as in a region downstream of *GCG*, we observed increased open chromatin in Jamaican fruit bat alpha cells (Fig. 5F, Supplementary Fig. 11i), although *GCG* was not differentially expressed between these species (Supplementary Fig. 11f). The *INS* promoter in Jamaican fruit bats is predicted to bind *NR2C2* and *USF2* (Fig. 5E). *NR2C2* is known to have a role in beta cell regulation and mice lacking *NR2C2* have hypoglycemia[163] and increased oxidative stress[164], whereas *USF2* expression is stimulated by high glucose and is known to control the synthesis of insulin[165-167]. There were many high-scoring motif occurrences for the *INS* promoter in big brown bats, including *GLI2*, which can serve as a transcriptional repressor in embryonic development[168], and transcriptional repressor *ZNF263*[169] (Fig. 5E). The *GCG* promoter in Jamaican fruit bats is predicted to bind *RREB1* (Fig. 5F), which regulates islet cell function[170]. The insectivore *GCG* promoter is predicted to bind *NRL* (Supplementary Fig. 11i), a motif that was also identified in human *GCG*[171]. The Jamaican fruit bat peak downstream of *GCG* is predicted to bind *FOXA1* (Fig. 5F), which regulates alpha cell differentiation, glucagon synthesis and secretion[172]. The homologous sequence in big brown bats is predicted to bind *POU2F2* (Fig. 5F). The lack of differential expression of *GCG* between bats with different diets may be attributed to other *cis*- or *trans*-acting regulators. Of note, however, our scATAC-seq analyses found Jamaican fruit bats to have both *GCG* and *INS* proximal and distal *cis*-regulatory regions that show a more open chromatin state compared to big brown bats (Fig. 5E, F, Supplementary Fig. 11h, i).

## Discussion

Using integrative single-cell sequencing, we characterized the cell populations, transcriptomes, and regulomes of insectivorous big brown bats and Jamaican fruit bat kidneys and pancreases in a single-cell manner. We identified major cell-types in the kidneys and pancreases of these bats, dissected the transcriptional and regulatory differences between them, and validated several cell composition findings with immunofluorescence. For the Jamaican fruit bat kidney, we found a reduction in loop of Henle cells, combined with a loss of urine-concentrating transporters. We observed an expansion of collecting duct cells, combined with upregulation of sodium reabsorption-potassium secretion genes, and genomic enrichment for disease phenotypes (Fig. 6). For the Jamaican fruit bat pancreas, we found an expansion of beta and alpha cells, accompanied by a reduction in acinar cells, and several genes and genomic regions associated with insulin secretion and signaling (Fig. 6). Combined, our work provides a cellular and molecular blueprint of a frugivorous bat species

and can inform potential therapeutic targets for human disease, particularly hypertension, hyperkalemia and diabetes.

Our unbiased and comprehensive analysis of the cell-type compositions that distinguish these insectivorous and frugivorous mammals provides several important insights. For the kidney, we find that the medullary and cortical differences observed between Jamaican fruit bat and big brown bat kidneys[16,17,63] are due to specific nephron composition differences in DTL, TAL, connecting tubules, principal cells, type A intercalated cells, type B intercalated cells, and proximal tubules-like cells (Fig. 6). As loop of Henle cells, DTL and TAL, are responsible for urine concentration and water recovery, and fruit bats get a substantial amount of water from fruit, fruit bats likely do not need as much of a structure for preserving water while excreting waste. A reduced renal medulla in response to water availability in diet or climate has also been observed in birds[173-175]. The greater abundance of connecting tubules and collecting duct cells in Jamaican fruit bats highlights nephron restructuring in response to high potassium and bicarbonate and low sodium and acid, favoring a larger ASDN. For the pancreas, we found that the large amount of endocrine tissue observed in fruit bats[18,19] is due to greater beta and alpha cell abundances, relative to insectivorous bats, and a compensatory reduction in exocrine tissue, specifically in acinar cells (Fig. 6). These cell composition differences likely contribute to the unique ability of fruit bats to lower their blood sugar rapidly, even faster than insectivorous bats[3,22-24]. Taken together, our integrative single-cell analysis identified several cell composition differences between the Jamaican fruit bats and big brown bats and provides a cellular-level detailed catalog of previously observed morphological frugivorous kidney and pancreas features.

Prior to our study, identifying potential molecular adaptations to frugivory was mainly restricted to molecular evolutionary analyses of specific genes and comparative genomics[3,20,21,34-40,87]. Here, we were able to use scRNA-seq to systematically identify gene expression differences between species in an unbiased manner. Our gene expression analyses identified numerous molecular differences between these frugivorous and insectivorous bats that could be associated and vital for fruit specialization in mammals. In the kidney, the dilute urine observed in fruit bats[31] is likely attributed to decreased expression of key urine-concentrating transporters, *STK39*, *OXSR1*, *KCNJ1*, *SLC9A3*, within the TAL (Fig. 6). We also found that the Jamaican fruit bat kidney exhibits gene expression changes to stimulate sodium reabsorption and potassium excretion, such as *KCNMA1*, *WNK1*, and *WNK4*, resembling an activated RAAS, which supports low sodium, high potassium dietary specialization (Fig. 6). Some genes that were lost in OWFB genomes, like kidney transporters *SLC22A6* and *SLC22A12*, have been hypothesized to be adaptive for frugivory[20], and our data allowed us to determine whether these hypotheses are lineage-specific (loss of *SLC22A6* and *SLC22A12* is specific to OWFBs). In the pancreas, the increased sensitivity to insulin and glucose observed in fruit bats[18,19] likely involves many genes that are associated with insulin secretion and signaling, such as *PTPRD* and *ABCC8* (Fig. 6). Moreover, we discovered a unique connection in myo-inositol transport and metabolism from our data and previous gene expression analyses in the same fruit bat species[133] (Fig. 6).

Our scATAC-seq datasets allowed us to identify gene regulatory elements and TFBSs that could be involved in mammalian frugivory. For the kidney, we found regulatory elements for many differentially expressed and diabetes-associated genes, including *KLK1*, which demonstrated cell-type-specific expression in bats with both promoter and enhancer accessibility, and *PCK1*, which was also shown to have differential chromatin accessibility downstream of the gene body in human diabetic proximal tubules[94]. Jamaican fruit bat renal epithelial cells were enriched for diabetes-associated motifs, such as *FOXO3*[94], and were highly enriched for human kidney phenotypes and for RAAS in particular, whereas insectivore renal epithelial cells were enriched

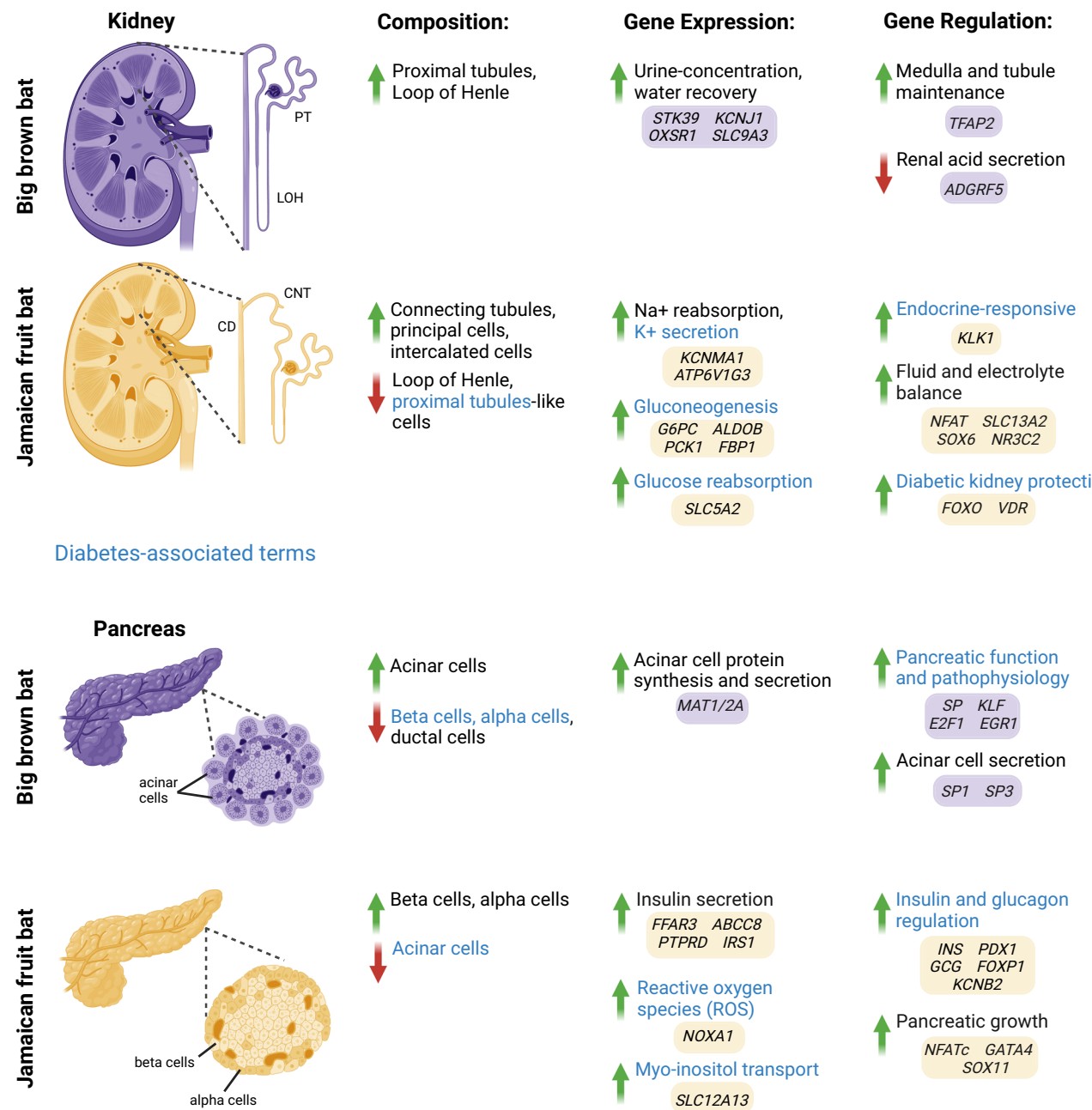

**Fig. 6 | Summary of cell composition, gene expression and gene regulation differences between the big brown bat and Jamaican fruit bat kidney and** pancreas and how it relates to human diabetes. PT proximal tubules, LOH loop of Henle, CD collecting duct, CNT connecting tubules (created with BioRender.com).

for RAAS downregulation with *EGR1* TFBS. For the pancreas, we found *INS* and *GCG* promoters to be highly accessible in Jamaican fruit bat beta and alpha cells, respectively. As frugivorous bats are known to have remarkable blood sugar regulation[3,22–24], promoter accessibility of *INS* and *GCG* may play a role. Both Jamaican fruit bats and big brown bat endocrine cells were enriched for TFBS of TFs involved in glucose homeostasis, such as *FOXP2* in Jamaican fruit bats and *EGR1* in big brown bats. Jamaican fruit bat exocrine cells had motif enrichment of pancreatic growth TFs like *NFATc1-3* and GATA4, whereas big brown bat exocrine cells had motif enrichment of acinar cell regulators *SP1* and *SP3*. As metabolites like lipids and carbohydrates regulate transcription through macronutrient-sensing TFs[176], the regulatory datasets we generated in this study offer the opportunity to investigate regulatory element evolution in response to dietary specialization.

Our joint scRNA-seq and scATAC-seq datasets allowed us to explore GRNs that could be pertinent for frugivorous bats. Fruit bat

kidney GRNs were enriched for glucose, acid-base, and electrolyte homeostasis, with *SLC13A2* displaying high centrality in the frugivore kidney. RAAS gene *NR3C2* displayed very contrasting networks between Jamaican fruit bat and big brown bat kidneys, with frugivores having many glucose- and diabetes-associated genes such as *PCK1* and *TOX*. In contrast to the kidney, pancreas GRNs showed similar pathway enrichment between bats, but there are clear differences in the genes involved. For example, *KCNB2* displays high centrality in Jamaican fruit bats whereas *KCNJ6* displays high centrality in big brown bats.

The limitations of our study include our sample sizes, genome assemblies annotations, and the lack of functional genomic datasets for bats to have known markers in bat kidneys and pancreases. Nonetheless, known markers in kidneys and pancreases of humans and mice were sufficient to identify major cell-types in the bat kidneys and pancreases, as has previously been reported[18,19,28], and known human and mouse kidney markers were identified in single-cell data for

horseshoe bats[49]. However, using human and mouse markers prevented us from identifying novel cell-types in bats. The smaller Jamaican fruit bat sample size for the pancreas prevented us from detecting significant differences in cell composition, but we were still able to identify differences between bats with high confidence. Some of the differences observed between the two bats may be caused by differences in the ability of single-cell technology to capture expressed genes, and orthogonal assays would be necessary to determine technology biases. Nonetheless, there appears to only be tissue bias, which is a common occurrence with single-cell technology[177,178]. Future studies may utilize spatial transcriptomics to compare cellular architecture and structures. We were also limited in analysis tools for non-model organism genomes. High-quality genomes are needed for deeper analyses, as they can provide better gene annotations and transcriptional isoforms and improve connection of gene regulatory elements to their target genes. Furthermore, novel genome annotations are needed to increase the number of shared features that can be recognized between species and to examine more genes in relation to humans and human disease. Additional bat genomes, functional genomic databases, and more single-cell datasets from other insectivorous and frugivorous mammals will further determine whether our findings are specific to the species we investigated here, to NWFBs, and/or to other frugivorous mammals. Our study may only reflect species differences between the Jamaican fruit bat and the big brown bat. Of note, our immunofluorescence results bolster previous hypotheses that the morphological differences between Jamaican fruit bats and big brown bats are likely diet-driven[16–19,28]. Future studies with similar tissues could further investigate the differences between fasted and fed states within each bat species at various time points and understand developmental mechanisms for cell composition differences between the kidneys and pancreases of frugivorous and insectivorous bats. In addition, our study only focused on two organ types (kidney and pancreas) and thus could not uncover tissue adaptations in other organs that might be associated with frugivory.

Bats have been viewed as a model for diabetes research due to endocrine tissue differences between bat species and their unique blood sugar regulation. Our detailed analysis of a mammalian frugivorous kidney provides several insights regarding the cellular and molecular mechanisms required for a diet with high sugar content. The Jamaican fruit bat kidney exhibits many diabetic features, including decreased proximal tubules, upregulation of gluconeogenesis, glucose reabsorption, and potassium secretion genes, and corresponding diabetes-associated motifs, including *FOXO3* which was also enriched in T2D renal tubules[94] (Fig. 6). Many of these kidney traits correspond with what we found in the Jamaican fruit bat pancreas, such as increased transport and need for myo-inositol and motif and expression enrichment of diabetes-associated genes (Fig. 6). We also provide detailed analyses of bat pancreases, in which we documented differential expression of many signature genes in human T2D beta cells and differential enrichment of diabetes-associated motifs in endocrine and exocrine cells (Fig. 6). Together, our integrative single-cell study indicates that Jamaican fruit bats evolved many diabetic-like features to deal with their diets but evolved protective mechanisms that prevent disease, such as upregulation of *KLK1* in type A intercalated cells in the kidney, which protects against diabetic tissue damage[74], and downregulation of *IRS1* in beta cells in the pancreas, which causes hyperinsulinism but not diabetes[144]. Our study provides a unique perspective for human disease therapeutic development by revealing cellular and molecular differences to high sugar, potassium, and bicarbonate levels and to low amounts of protein, sodium, and calcium. *Cis*-regulation therapy (CRT)[179], for example, can take advantage of the genes and regulatory elements identified here for metabolic disease treatment. Moreover, our cell-type-specific data provides insight for designing localized treatments in heterogeneous tissues.

## Methods

Our research complies with all relevant ethical regulations. All experimental procedures were approved under IACUC protocols (NEOMED IACUC #20-06-721, CSU IACUC #1034) and the AMNH Institutional Animal Care and Use Committee (AMNH IACUC-20191212). Capture and sampling were conducted under Belize Institute of Archaeology Permit IA/S/5/6/21(01) and Belize Forest Department Permit FD/WL/1/21(16), and samples were exported under Belize Forest Department permit FD/WL/7/22(08).

### Bat samples and dietary treatment
For Chromium single-cell Multiome ATAC + Gene Expression experiments, kidneys and pancreases were obtained from four adult male *Artibeus jamaicensis* fruit-eating bats (Colorado State University [CSU]) and five adult male *Eptesicus fuscus* insect-eating bats (6-7 years old; Northeast Ohio Medical University [NEOMED]). We used males from both species due to availability and ability to carry sex matched comparisons. Jamaican fruit bats (*Artibeus jamaicensis*) were housed in a 65^2 m free-flight room and provided fresh food and water daily. The food is provided ad libitum and composed of watermelon, cantaloupe, banana, and honeydew. Protein supplement (Body Fortress Super Advanced Whey Protein Powder, Gluten-Free Vanilla) at approximately 20% of the weight of the fruit (average about 600 g per day). Some of the fruit is hung on wall-mounted skewers to provide enrichment. Landscape fabric is draped on the walls and ceiling as a roosting substrate. Bats are monitored daily by CSU Laboratory Animal Resources staff and provided veterinary care as needed. All Jamaican fruit bats were fasted overnight during their night cycle (approximately 12 hours), and fasted bats were removed from their enclosures before feeding for euthanasia. Fed Jamaican fruit bats were provided unlimited cantaloupe and banana and were euthanized 30 minutes later. Tissues were harvested immediately for flash-freezing (May 2021) and stored at −80 Celsius until nuclei isolation. All big brown bats (*Eptesicus fuscus)* were housed as a colony in a large, indoor flight enclosure at NEOMED[180] with mesh siding on one wall, and baskets and towels for roosting. The bats consume meal worms and water *ad libitum*. Dishes of water and meal worms were replaced daily, and the enclosure was cleaned regularly. To support the health of the bats, the meal worms are fed a diet of oat bran, ground monkey and rodent diet meals, powdered milk, a skin and coat formula, apples, sweet potatoes, and a multivitamin. Because big brown bats feed on mealworms (*Tenebrio molitor*), we increased the fruit content of these mealworms to create a fruit treatment for these insect-eating bats. These mealworms were fed a modified fruit diet for four days of sweet potatoes, apples, mango, and cane sugar. Fasted big brown bats were removed from their enclosures before feeding for euthanasia, and three fed big brown bats were fed an unlimited supply of fruit-enriched mealworms and were euthanized 30 minutes later. Tissues were harvested immediately for flash-freezing (all big brown bats were euthanized before the winter (September 2020) to exclude hibernation effects) and stored at −80 Celsius until nuclei isolation. All experimental procedures were approved under IACUC protocols (NEOMED IACUC #20-06-721, CSU IACUC #1034).

For immunofluorescence experiments, kidneys and pancreases were obtained from 3 adult male big brown bats of unknown age (June 2022; NEOMED IACUC #20-06-721), 2 adult male Jamaican fruit bats (August 2022; CSU IACUC #1034), and 1 adult female Jamaican fruit bat captured on an American Museum of Natural History (AMNH) field expedition to the Lamanai Archaeological Reserve in Orange Walk District of Belize in November 2021. The individual sampled (field number BZ701, voucher number AMNH.Mammalogy.281145) was caught in a ground-level mist net set near the High Temple at Lamanai (17.76750 N, 88.65207 W) on 17 November 2021. The bat was subjected to minimal handling after capture, and it was held in a clean cloth bag after capture as per best practices for field containment of bats. After

species identification, the individual was euthanized humanely by isoflurane inhalation the same night it was captured. Capture and sampling were conducted under Belize Institute of Archaeology Permit IA/S/5/6/21(01) and Belize Forest Department Permit FD/WL/1/21(16), and samples were exported under Belize Forest Department permit FD/WL/7/22(08). All work was conducted with approval by the AMNH Institutional Animal Care and Use Committee (AMNH IACUC-20191212). Tissues were removed from the subject individual immediately following euthanasia and were flash-frozen in a liquid nitrogen dry shipper, with the cold chain maintained from field to museum to laboratory.

### Single-cell multiome ATAC and gene expression sample processing and sequencing

For Chromium single-cell Multiome ATAC + Gene Expression experiments, we followed the manufacturer's protocols for complex tissues (10x Genomics: CG000375, CG000338). Nuclei were sorted on a FACS Aria II (Becton Dickinson) using a capture target of 10,000 nuclei per sample to prepare libraries. ATAC libraries and GEX libraries were pooled together by tissue. Tissue-specific libraries were sequenced PE150 on two lanes of NovaSeq 6000 (Illumina) (Supplementary Data 1). The pancreas of a fed Jamaican fruit bat was not used due to low quality sequencing data.

### Bat genome modifications for joint scRNA and scATAC analysis

Jamaican fruit bat and big brown bat genomes and annotations were downloaded from NCBI [GenBank Assembly Accession: GCA_014825515.1 (*artJam2*), GCA_000308155.1 (*eptFus1*)]. Scaffolds smaller than 50 kb for each genome were removed. Gene information in annotation files was collapsed so that each gene was represented by a single "exon" transcript. A mitochondrial genome was generated for the big brown bat using GetOrganelle version 1.7.5[44], with default parameters for animal mitochondria assembly, and the output fasta was input into MITOS[181], with vertebrate genetic code, to create the corresponding gtf file (GenBank TPA: BK063052). Each species annotation was modified with one-to-one orthologue IDs created from Orthofinder[45] (Supplementary Data 13), which takes proteomes as input and assigned 89.9% of the total bat genes.

### Individual sample scRNA-seq and scATAC-seq processing and quality control (QC)

A total of 15 scRNA and scATAC FASTQs were input into cellranger-arc 2.0 (10x Genomics) whereby raw feature barcode matrices were generated with "count". ATAC matrices were loaded as an object into Seurat version 4.0[46]. A common peak set was created within a species with R package GenomicRanges 1.50.2[47] "reduce" (Supplementary Fig. 1d). RNA matrices were then combined with ATAC matrices using only cells that overlapped the existing object. Cells were filtered for mitochondrial percentage <25%; ATAC peak count 500 <x < 100,000; RNA count 200 <x < 25,000; nucleosome signal <2; TSS enrichment >1 (terms defined by ENCODE[182]).

### Joint scRNA-seq and scATAC-seq bioinformatics workflow within species

Using post-QC Seurat objects as generated above, SCTransform was used to normalize each sample with mitochondrial percentage as a regressed variable and to find the top 3000 variable features from each sample. Species replicates were then merged. To integrate RNA (SCT) across all samples within a species, 3000 repeatedly variable features across each sample were selected with "SelectIntegrationFeatures", and Harmony was run on the PCA dimensions 1:30, removing original sample identification as a variable. To integrate ATAC across all samples within a species, peaks were normalized with Signac function "RunTFIDF", the most frequently observed features were identified with Signac version 1.8.0[47] function "FindTopFeatures" with

"min.cutoff = 'q0'", and Harmony was run on the singular value decomposition (SVD) dimensions 2:30, removing original sample identification as a variable. To integrate the integrated RNA and ATAC modalities, weighted nearest neighbor analysis[46] was applied on the Harmony reductions using 30 dimensions for each modality. Seurat function "FindClusters" was used with SLM algorithm and 0.6 resolution to identify clusters. Seurat function "FindAllMarkers" was used to identify cluster-specific markers and manually assign clusters according to shared markers to known mouse and human cell-types.

### Joint scRNA-seq and scATAC-seq bioinformatics workflow across species

Using post-QC Seurat objects mentioned above, gene activity matrices were created for each sample with the Seurat function "GeneActivity", and ATAC peaks were removed from each Seurat object. R package SCTransform[183] was used to normalize each sample with mitochondrial percentage as a regressed variable and to find the top 3000 variable features from each sample. Samples of the same species were then merged while retaining SCT matrices. Merged gene activity matrices were then log-normalized with the Seurat function "NormalizeData" with "scale.factor" set to the median of the gene activity counts. Two-thousand variable features were detected for normalized gene activity scores for each species with Seurat function "FindVariableFeatures" (vst method). Species were then merged. To integrate RNA (SCT) across species, we used Seurat function "SelectIntegrationFeatures" to select the 3000 top scoring variable features across all samples and manually added the selected variable features to the merged samples, and Harmony[48] was run on the principal component analysis (PCA) dimensions 1:30 with 20 maximum iterations and removing original sample and species variables. To integrate gene activity scores across species, we used Seurat function "SelectIntegrationFeatures" to select the 2000 top scoring variable features across all samples and manually added the selected variable features to the merged samples. The features were scaled with Seurat function "ScaleData", and Harmony was run on the PCA dimensions 1:30, removing original sample identification and species variables. To integrate the RNA and gene activity score modalities, weighted nearest neighbor analysis[46] was applied on the Harmony reductions using 30 dimensions for each modality. Seurat function "FindClusters" was used with SLM algorithm and 0.6 resolution to identify clusters, and "FindSubClusters" was used with original Louvain algorithm and 0.1 resolution to identify subclusters in pancreatic acinar cells. Seurat function "FindAllMarkers" was used to identify cluster-specific markers and manually assign clusters according to shared canonical markers to known mouse and human cell-types (Supplementary Fig. 1d).

### Cell composition analyses

We calculated cell-type proportions from each sample and compared proportion percentages between the species by calculating significance with the Wilcoxon signed-rank test. The resulting boxplots for proportion percentages for each species and p-values were visualized with R package ggplot2 version 3.4.0[184]. For calculating species or condition associations per cell, we used R package CNA version 0.0.99[56]. For species associations, we used species variables as testing variables and used the neighbor graph generated from Seurat's Weighted Nearest Neighbor Analysis from above. For condition associations, we used condition variables as testing variables and regressed out effects from species variables. For mosaic plots and calculation of p-values with pearson residuals, we used R package VCD version 1.4.10[185] with species information and cell-type annotations.

### Automated cell-type annotations with human and mouse references and comparison

For the automated cell-type annotation method using human and mouse single-cell references, we used R package Azimuth version

0.4.6[46] with built-in Azimuth human kidney dataset[46,60], mouse P0 and adult kidney[58], and human adult pancreas[123]. For reference datasets with several levels of annotations, the most major cell-types were selected. To compare the similarity between manual annotations and automated annotations, we calculated the overlap coefficient between cells from each cell-type with different annotation methods. The resulting similarity heatmaps were visualized with R package pheatmap version 1.0.12[186].

## Immunofluorescence staining, imaging and analysis

Flash-frozen samples were embedded in Tissue-Tek OCT Compound (Sakura Finetek, Torrance, Ca, USA), cut as 14uM cryo-sections, and fixed in 4% paraformaldehyde for 10 minutes at room temperature. Sections were rinsed in Phosphate Buffered Saline (PBS) three times and blocked in a humidity chamber in 3% normal donkey serum (Sigma-Aldrich, 566460) and 0.3% Triton X-100 (Sigma-Aldrich, T8787) in PBS for one hour at room temperature. Sections were then incubated overnight at 4 degrees Celsius in a humidity chamber protected from light with one of the following primary antibodies from Thermo Scientific™ at respective dilutions in blocking buffer: *SLC12A1* (18970-1-AP; 1:100), *AQP2* (PA5-78808; 1:100), *SLC26A4* (PA5-115911; 1:100), *INS* (15848-1-AP; 1:500), *GCG* (15954-1-AP; 1:500). Epitope matching can be found in Supplementary Table 1. Sections were washed three times with PBS and incubated in a humidity chamber protected from light for two hours at room temperature with a secondary antibody Donkey Anti-Rabbit IgG NorthernLights™ NL557-conjugated Antibody (Biotechne, NL004; 1:200 in blocking buffer). Sections were rinsed with PBS three times and either incubated with Hoechst 33342 Solution (20 mM) (Thermo Scientific™, 62249; 1:10000) for 5 minutes before being mounted with ProLong™ Diamond Antifade Mountant (Invitrogen™, P36970) or mounted with ProLong™ Diamond Antifade Mountant with DAPI (Invitrogen™, P36962). Tissue sections were imaged on a CSU-W1 Spinning Disk/High Speed Widefield microscope with a Plan-Apochromat 20x objective and the Andor Zyla 4.2 sCMOS camera for confocal imaging or the Andor DU-888 EMCCD camera at the UCSF Center for Advanced Light Microscopy (CALM). On a section of tissue, ten random z-stack images (2-22 sections at .92uM thickness; image bit depth 16) were collected using the same imaging specifications respective to the target antibody. Fluorescence illumination was kept to a minimum to avoid photobleaching. All images shown are single z-sections processed with ImageJ on the FIJI platform[187] using the same processing parameters respective to the target antibody.

Tissue section nuclei count and total antibody fluorescence intensity were measured for every z-section for each image with CellProfiler version 4.2.4[188] using the same detection framework respective to the organ and to the target antibody. A fluorescence intensity threshold (global threshold strategy, Otsu thresholding method, three-class thresholding) was created for each antibody to reduce background signal. Nuclei counts and total antibody fluorescence intensities (arbitrary units [AU]) were summed across each image z-stack. The sum total antibody intensity was normalized to the sum nuclei count to obtain total antibody intensity (AU)/nuclei for each of the 10 images per tissue section. These 10 normalizations were then averaged to get the tissue section average of total antibody intensity (AU)/nuclei. To obtain the species average of total antibody intensity (AU)/nuclei, we averaged the tissue section averages by species. Results of immunofluorescence experiments represent mean ± standard error of the mean (SEM) derived from 3 individual insectivorous bats and 3 frugivorous bats ($n = 3$/phenotype). *P* values were calculated using a mixed effects model with restricted maximum likelihood and Satterthwaite approximation for degrees of freedom.

## Differential expression and gene set enrichment analyses

We used Seurat function "FindMarkers" for the calculation of differentially expressed genes. We used a threshold of average log base2 fold change > 0.25, adjusted *p*-values < 0.01, and minimum detected gene fraction (min.pct) > 0.25. For DEGs between species, we additionally filtered out genes that are not expressed in both species. The differentially expressed genes were visualized with volcano plots using R package ggplot2 version 3.4.0[184]. We visualized selected genes on UMAP embeddings with Seurat function "FeaturePlot" and "VlnPlot" for violin plots. For gene set enrichment analysis using DEGs, we selected top genes with smaller counts of 100 or all DEG counts for each condition. We used R package EnrichR version 3.0[189] for stated pathway databases and *p*-value calculations. The gene set enrichment results were visualized with ggplot2 with adjusted *p*-values calculated with EnrichR.

## Bat-human gene and genome comparison analyses

TOGA version 1.13[51] human (*hg38*) one-to-one ortholog, transcript, and gene coordinate predictions were downloaded for the insectivorous bat genome (*eptFus1*) and fruit bat genome (*artJam2*) (last downloaded July 1st, 2023). Using differentially expressed genes identified in bat kidneys and pancreases, we determined how many had at least one TOGA ortholog prediction between human and each bat. For transcript predictions, we determined how many of the differentially expressed genes were predicted to have at least one intact or partially intact gene transcript. For gene coordinates, we mapped predictions from the annotated chain to its paired scaffold.

To calculate pairwise synteny between each bat genome (>50 kb scaffolds) and human (*hg38*) and mouse (*mm39*) and human, we ran MCscan[52] (Python version 1.3.6) with default parameters. To be considered a syntenic block, the block must include at least 4 gene pairs[52].

## Peak calling and enriched motif and GREAT analyses

ATAC data was added back into the integrated Seurat objects (Supplementary Fig. 1d) following separation by species via the Seurat function "SplitObject()". Peaks were called for each cell-type by MACS2[96] with "-nomodel -extsize 200 -shift −100" and effective genome size respective to each bat genome. Cell-type-specific peaks were associated genes and human and mouse phenotypes with the Genomic Regions Enrichment of Annotations Tool (GREAT) version 4.0.4[98]. GREAT only takes human or mouse coordinates for input, therefore big brown bat peaks were lifted to the human genome *hg38*[97] using UCSC Genome Bioinformatics Group tools chainSwap and liftOver[190] with -minMatch = .1 and chain file "hg38.eptFus1.all.chain". To lift Jamaican fruit bat peaks to *hg38* for input into GREAT, a chain file was created between *hg38* chromosomes 1-22 and tandem repeat-masked Jamaican fruit bat genome via Lastz 1.04[191] and UCSC Genome Bioinformatics Group tools trfBig, mafToPsl and pslToChain[190]. GREAT defines a basal regulatory domain for a gene as 5 kb upstream and 1 kb downstream of the gene's TSS, regardless of other nearby genes, and extends this domain in both directions to the nearest gene's basal domain, but no more than 1000 kb in one direction. We viewed outputs from "Significant By Region-based Binomial" view.

Differentially enriched TFBSs from cell-type-specific peaks were estimated with AME version 1.60[99], setting each time the other species' cell-type-specific peaks as control sequences and using default parameters with JASPAR2022 CORE vertebrates as motif reference. We ranked enriched motifs based on their adjusted *p*-values and converted the ranks to percentile ranks in order to compare results with different total motif counts. In order to compare motif enrichments, we selected motifs that are ranked as the top 10% from each species and cell type combination and removed any motifs that are unique to one combination. TFBS prediction on regulatory sequences was executed with FIMO version 1.60[192] and filtered for *p*-value < 0.0000006 and *q*-value < 0.0003. For AME and FIMO motif inputs, we used the JASPAR 2022 core redundant dataset for vertebrates[193].

## Multi-omics gene regulatory network (GRN) analysis

Versions of insectivorous and frugivorous bat genomes were manually constructed for BSgenome[194] (version 1.68.0), and these versions were used as reference genomes with multi-omic data into Pando (version 1.0.5)[111] using default TF and TF motif lists built in Pando. We built GRNs separately for each species and each tissue. For inferring GRNs with infer_grn() function, we chose the "Signac" method for building peak-to-gene connections with variable genes identified for each species. To compare inferred GRNs between two species for the same organ, we calculated strength centrality with igraph[195] (version 1.5.0) function "strength" and ranked each node with percentile ranks for each species. To prioritize nodes that have both high centrality and specificity for each species, we selected nodes with top 100 or higher than top 50% centrality percentile ranks for one species and lower than top 25% centrality percentile ranks for the other species. With the species-specific top nodes, we ran pathway analysis with the following pathways: BioPlanet (2019)[196], GO Biological Process (2023)[69,70], Reactome (2022)[197], and WikiPathway Human (2021)[198].

## Statistics and reproducibility

No statistical method was used to predetermine the sample size. Samples were randomly selected. Single-cell profiling of all adult male tissues was done with two fasted and two fed bats, with the exception of *A. jamaicensis* pancreases due to low sequencing quality from one fed pancreas. Immunofluorescence experiments were carried out with three adult male *E. fuscus* bats and two adult male and one adult female *A. jamaicensis* bats. The investigators were not blinded to allocation during experiments and outcome assessment.

The calculation methods applied are detailed in respective figure legends. An unpaired two-sided Wilcoxon rank-sum test without correction was used for the comparison of cell-type composition between two species. We used a mixed effects model (two-sided) with restricted maximum likelihood and Satterthwaite approximation for degrees of freedom for the calculation of *p*-values for quantification of immunofluorescence. One-sided Fisher's exact test with Benjamini-Hochberg correction was used for pathway analysis. For differentially expressed genes and comparison of gene expression with violin plots, we used a two-sided Wilcoxon rank-sum test with the Bonferroni correction and any *p*-values lower than 1e-300 were considered 0. The numbers of biologically independent samples and experiments examined in this manuscript can be found in respective figure legends and figures.

## Reporting summary

Further information on research design is available in the Nature Portfolio Reporting Summary linked to this article.

## Data availability

Raw 10X multiome sequencing data generated in this study were deposited at NCBI BioProject under accession code PRJNA916224. Big brown bat mitochondrial genome nucleotide sequence data reported are available in the Third Party Annotation Section of the DDBJ/ENA/GenBank databases under the accession number TPA: BK063052. Human genome (*hg38*) sequence used for Jamaican fruit bat pairwise alignment can be found at GenBank:GCA_000001405.19. Mouse genome (*mm39*) and human genome (*hg38*) sequences used for MCscan can be found at GenBank: GCA_000001635.9 and GCA_000001405.29. All data are available in the main text or the supplementary materials. Source data are provided with this paper.

## Code availability

The code used in this manuscript are available on GitHub: https://github.com/netbiolab/multiome_bat/. These code can be cited with the following DOI identifier: https://doi.org/10.5281/zenodo.8254234[199].

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

## Acknowledgements

The authors thank Carolyn Ku at UCSF for assistance with bat genome alignments. The authors also thank all the 2021 Belize Batathon members for assistance with bat collection. Figures 1A–C, 4A, 6 and Supplementary Fig. 2 were created with BioRender.com. This work was funded in part by National Human Genome Research Institute grant R01HG012396 (WEG, NA), Korea Health Technology R&D Project, Korea Health Industry Development Institute (KHIDI), Ministry of Health & Welfare, Republic of Korea grant HI19C1344 (SB), California Institute for Regenerative Medicine (CIRM) Postdoctoral Fellowship (HPN), UCSF Vision Core shared resource of the NIH/NEI P30 EY002162 (YMK), UCSF CALM shared resource of the 1S10OD017993-01A1 (WEG), National Institute on Deafness and Other Communication Disorders of the U.S. Public Health Service Research grant R01 DC00937 (AG), Smithsonian National Museum of Natural History Peter Buck Postdoctoral Fellowship (MRI), National Institute of Allergy and Infectious Diseases (NIAID) grant R01AI134768 and R01AI140442 (TS), National Science Foundation (NSF) grant 2020297257 (TS), and Evergrande Center startup funding (MH).

## Author contributions

W.E.G. and N.A. conceived the study. W.E.G., M.R.I., N.B.S. and N.A. conducted fieldwork. W.E.G., S.B., H.P.N., Y.M.K., I.G-S., M.H. and N.A. were involved in methodology. A.G., I.L., M.R.I., N.B.S., T.S., L.N.C., M.H. and N.A. were involved in project administration. W.E.G., S.B., H.P.N., R.B., S.F., N.K. and I.G-S. performed experiments. W.E.G. and S.B. were responsible for visualization. W.E.G., S.B., M.H. and N.A. wrote the manuscript with contributions from all authors. W.E.G., S.B., M.H. and N.A. acquired funding and supervised.

## Competing interests

N.A. is a cofounder and on the scientific advisory board of Regel Therapeutics. N.A. receives funding from BioMarin Pharmaceutical Incorporate. The other authors declare no competing interests.
