## [Peer Review File · Nature Communications]

Integrative single-cell characterization of a frugivorous and an insectivorous bat kidney and pancreasREVIEWER COMMENTS

Reviewer #1 (Remarks to the Author):

Gordon and colleagues present a single cell (RNA-seq and ATAC-seq) analysis of frugivory adaptations in the bat kidney and pancreas. This is an interesting timely study that will attract significant attention especially for human diabetes. Besides the valuable new insights, this work is a powerful resource for other scientists to deepen into the understanding of frugivorous adaptation in the process of evolution.

That said, as a resource, there are a number of important points and more in-depth analysis that will improve the usefulness of this dataset for the community. I do have the following concerns about this work:

1. The results of clustering are slightly unclear between clusters, which might be due to the existence of ambient RNA and doublets. Can the authors perform soup and doublet removal on the data to improve the clustering results?
2. The authors showed that the high similarity between their cell-type annotations and automated annotations in Fig.1h-i and Fig.3f-g. However, only three cell-types had higher correlations in Fig.1i, and the cell-type distribution in the UMAP of Fig.1h and Fig.1d seems quite different, for example, the proximal tubules and descending thin limbs. The authors need to provide an explanation about the accuracy of cell-type identification.
3. In the pancreas cell-type composition analysis, the composition of B cells and T cells were significantly different between two species. Is this a cause or a result of single nuclei dissociation or the true differences across species?
4. For the DEG analysis, the authors focused on the cell-type with different composition between insectivorous and frugivorous bat kidneys. Are there gene expression differences between cell-type with similar composition?
5. The authors converted bat peaks to human genome coordinates for downstream analysis. Over 60% bat peaks can align to human genome. For the other 30% bat-specific peak, it would be nice if the authors could further study the regulatory function, which will be important to explain the differences of glycometabolism between bat and human from an evolutionary point of view.
6. The authors indicated that fruit bats evolved many features similar to a human diabetic kidney with example of several motifs. Further system integration analysis between bat and human single-cell data is essential to improve the research significance of bat models.
7. In the gene regulatory part, the authors identified many TFs to explain the difference between species in kidney and pancreas. This is far from enough to understand the complex regulatory networks, especially for how these TFs regulate the different gene expression. A comprehensive multi-omics regulatory network needs to be constructed and further explored by bioinformatics tools such as Pando.

More minor comments:

1. The legend titles of Fig.2a, Fig.3a and Fig.5d were missed.
2. There are some issues with the formatting of reference index, for example, the 45.46 and 121.120 in the Methods.

Reviewer #2 (Remarks to the Author):

Overall Comments:

In general, the authors have done a great deal of work, and they have used the state-of-the-art technology called scRNA-seq and scATAC-seq. Their findings are interesting, but require further validations. The authors must check the potential biases as follows: 1) The observed difference between the two bats may be caused by the difference of the two bat species per se rather than dietary difference. The best way to rule out this possibility is to detect difference between one species of Old World fruit bat and one species of its insectivore relative, and find the shared difference between the authors' comparison and my suggested new comparison. 2) The observed difference between the two bats may be caused by the difference in the ability of single-cell technology to capture expressed genes; 3) The homology and functional consistency of cross-species genes need to be supported by sufficient data, especially the homology and functional consistency of protein-coding genes in the collinear regions of human and bat genomes needs to be proved.

Specific comments:

- 1) Page 5, line 13 of paragraph 2. What are you comparing with? Why is there an increase of the number of shared features?
- 2) Page 6, line 12-13 of paragraph 1. As far as I know, kidney scRNA or scATAC in bats is not the first. Please see Lv et al., 2022 (<https://doi.org/10.1038/s41597-022-01447-7>)
- 3) Page 6, the end of paragraph 1. Authors need to provide the consistency of cell type classification between the scRNA/scATAC data of replicate treatments from the same tissue in the same bat species.
- 4) Page 9. "For example, ATPase H⁺ transporting V1 subunit G3 (ATP6V1G3) was highly expressed in type B intercalated cells in fruit bats but not in insectivorous bats." Is this gene expressed in other tissues or cell types of insectivorous bats? It is necessary to exclude that it is not caused by the limitation that scRNA sequencing cannot comprehensively capture expressed genes?
- 5) Page 13. "we converted our bat peaks to human genome coordinates (hg38), converting 141,779 insectivorous and 124,845 frugivorous bat peaks to hg38, respectively. We then used the Genomic Regions Enrichment of Annotations Tool (GREAT) to characterize the target genes associated with these peaks." Are they orthologous genes in the syntenic regions of bat and human genomes? In other words, how could you be sure that these human genes can be found in the syntenic regions of bat genomes?
- 6) Page 14. "Our cell-type-specific regulatory profilings allowed us to investigate human disease variants that are difficult to study without cell-type-specific data, such as rs79687284 at the PROX1 locus. This variant is highly associated with T2D and was predicted to have regulatory function as an enhancer, but it did not demonstrate transcriptional activity in luciferase reporter assays in a pancreatic beta cell line that expresses PROX1". I believe the cross-species analysis is particularly interesting as it highlights the potential for bats to serve as an alternative model for studying diabetes in humans. Through GWAS analysis, you were able to locate a candidate site in PROX1. Your results suggest that PROX1 performs opposite functions in insectivorous bats and humans. You have identified that PROX1 has an enhancer regulatory function associated with the rs79687284 variant. To further investigate this, you need to evaluate the degree of homology between PROX1 in bats and humans. Specifically, does the bat site corresponding to the rs79687284 site match? Additionally, it is important to determine if bat PROX1 also has the regulatory function of an enhancer. Furthermore, what are the findings of the scRNA and scATAC analysis of this gene in frugivorous bat kidneys?

Reviewer #3 (Remarks to the Author):

Here, Gordon and colleagues perform a single nucleus multiomic comparison between fruit and insect bat kidneys and pancreas. Fruit bats have evolved to handle the higher sugar and potassium loads in their diets compared to insect bats but the precise cellular adaptations have not been characterized. The authors report increased collecting duct in fruit bats, which is consistent for important roles in potassium excretion and the fine tuning of sodium balance in this segment. In the pancreas, they report increased beta and alpha cells in fruit bats. They further find a diabetes-like signature in fruit bat proximal tubule including increased gluconeogenesis. A variety of individual genes and transcription factors and binding motif accessibility are analyzed with some validation by immunofluorescence as well.

Overall the study is well performed and reads clearly. The differing physiology is interesting. Limitations include the GWAS analysis and somewhat limited validation.

1. PT-like cells. These appear to be mostly characterized by decreased expression of PT anchor genes. Are there any upregulated genes compared to PT specific to this cluster? Single cell studies in human have identified other PT-like clusters with reduced expression of PT-defining genes, but also increased expression of others (Vcam1, CD133, CCL2, Prom1). Can the authors find any links between bat PT-like cells and these human PT cells that have variously been called "PT_VCAM1," "Maladaptive PT" or "Failed-repair PT"?
2. Does the GWAS analysis add anything? It does not make sense to me to take bat peaks and map human DM GWAS variants onto them, after lifting them over to the human genome. There is already plenty of human snATAC-data published onto which the GWAS hits can be mapped. Doing so using the bat peaks would seem to be less sensitive. At the least, the authors need to compare bat vs human single cell ATAC datasets and prove that the bat analysis is providing new/better information.
3. In the discussion, the authors comment on increased Nfat5 binding motifs in insectivores and interpret this to be consistent with the increased protein diet in insectivores consume state, because Nfat5 has been shown to regulate proteinuria. But in fact podocyte Nfat5 activity causes proteinuria so would not be protective in the setting of a high protein diet.

We want to thank the reviewers for their positive assessment of our work and for their extremely helpful comments, which have significantly improved our revised manuscript. Below is a point-by-point response to these comments with our answers in blue font. In addition, we marked our changes in blue font in the manuscript so they can be easily found.

Reviewer #1

Gordon and colleagues present a single cell (RNA-seq and ATAC-seq) analysis of frugivory adaptations in the bat kidney and pancreas. This is an interesting timely study that will attract significant attention especially for human diabetes. Besides the valuable new insights, this work is a powerful resource for other scientists to deepen into the understanding of frugivorous adaptation in the process of evolution.

That said, as a resource, there are a number of important points and more in-depth analysis that will improve the usefulness of this dataset for the community. I do have the following concerns about this work:

1. The results of clustering are slightly unclear between clusters, which might be due to the existence of ambient RNA and doublets. Can the authors perform soup and doublet removal on the data to improve the clustering results?

We thank the reviewer for this suggestion. Currently, there is no standard practice for joint scRNA and scATAC data on filtering ambient RNA and/or doublets, as some of these methods often filter out valid single cells. We followed the most current quality control methods (accessed 2022) from Signac¹ “Joint RNA and ATAC analysis: 10x multiomic” and Seurat² “WNN analysis of 10x Multiome, RNA+ATAC”. To address the reviewers’ suggestion, we ran SoupX³ and Scrublet⁴. For SoupX, we calculated contamination scores (ρ) of pre-QC and post-QC samples. Most of the samples did not have high contamination scores (around 10% is common <https://github.com/constantAmateur/SoupX>) even before QC, and contamination scores decreased for post-QC samples, except for several cases where contamination scores were already very low (**Fig R1**). For Scrublet, we calculated the doublet scores for post-QC samples. Most of the samples had fewer than 10 doublets, and the distribution of the doublet scores shows very low scores for most of the cells (**Fig R1**). These results together demonstrate that sufficient quality control methods were applied and the clustering results are not heavily impacted by ambient RNA and doublets.

Scrublet Doublet score

Figure R1: Quality control metrics for SoupX and Scrublet. (Top) Mean quality scores per sample and (Bottom) Scrublet score distributions. The vertical line indicates which cells were excluded.

2. The authors showed that the high similarity between their cell-type annotations and automated annotations in Fig.1h-i and Fig.3f-g. However, only three cell-types had higher correlations in Fig.1i, and the cell-type distribution in the UMAP of Fig.1h and Fig.1d seems quite different, for example, the proximal

tubules and descending thin limbs. The authors need to provide an explanation about the accuracy of cell-type identification.

The reviewer makes a good point that some of the Jaccard index scores are quite low and therefore accuracy of the cell-type identification could be low as well. There are several potential reasons for this:

1) The Jaccard index is affected by comparing groups of different sizes (see #2 below) and therefore tends to be smaller. Since each study we used for reference data has slightly different levels of cell-type identification, it is expected that the Jaccard index scores are not as high as may be expected for some cell-types. For example, in the kidney, we identified type A and B intercalated cells, but adult human kidney Azimuth reference data^{5,6} only identified “intercalated cells”. In the pancreas, the adult pancreas reference data⁷ did not identify immune cell sub-types (only “macrophages”), which might explain our bat immune cell sub-types mixing with the human reference “acinar cells”.

2) Differences in study sample processing and sample qualities, leading to differences in annotated compositions.

3) Differences in cell-type identification can be caused by cell-type and marker differences among human, mouse, and bat, and there are no known markers for bat pancreases and only one dataset for bat kidneys⁸.

4) Differences between gene names in each genome annotation. For example, the gene *SEPTIN1* in the human annotation is named *SEPT1* in the bat annotation, leading to a mismatch in similarity.

Since it is difficult to overcome cell-type identification differences caused by points #3 and #4, we chose to use overlap coefficients to address methodological differences among studies and calculate similarity between our manual annotations and automated annotations with the previous reference studies. We updated the manuscript accordingly (**Fig. 1i, Fig. 4g, Extended Data Fig. 6b**; see in Methods ‘Automated cell-type annotations with human and mouse references and comparison’ section). The overlap coefficients demonstrate that the major cell-types in human and mouse correlate well with cell-types identified in bat.

Here is an example of the difference between Jaccard index and overlap coefficients for kidney intercalated cells: For Azimuth kidney reference-based result, which only annotates “intercalated cells” (IC), whereas we identified types A (ICA) & B (ICB), the Jaccard indexes are: ICA-IC: 0.36 & ICB-IC: 0.42 & combined IC-IC: 0.74; and the overlap coefficients are: ICA-IC: 0.87 & ICB-IC: 0.82 & combined IC-IC: 0.85.

3. In the pancreas cell-type composition analysis, the composition of B cells and T cells were significantly different between two species. Is this a cause or a result of single nuclei dissociation or the true differences across species?

The significant difference in the composition of B cells and T cells in the pancreas cell-type composition analysis, detected by Chi-square tests of independence on Pearson residuals (**Extended Data Fig.10f**), is a result of one of the bats (fed fruit bat, J4) used for this study. We have added the figure below and the following text to the manuscript:

“The overwhelming majority of immune cells were found in only one Jamaican fruit bat (**Extended Data Fig. 10a,b**). Due to this immune cell bias and our low sample size, we focused only on non-immune cell types in our subsequent analyses.”

Figure R2: Cell type distribution across samples.

4. For the DEG analysis, the authors focused on the cell-type with different composition between insectivorous and frugivorous bat kidneys. Are there gene expression differences between cell-type with similar composition?

There are gene expression differences between cell-types with similar compositions across bat kidneys, most notably in proximal tubules. There was no significant difference in proximal tubule presence between bats in both Wilcoxon rank-sum tests (**Fig. 1g**) and Chi-square tests of independence on Pearson residuals (**Extended Data Fig.6f**), and we found gluconeogenesis and glucose reabsorption genes to be upregulated in fruit bat proximal tubules (**Fig. 2j-k**). Ultimately, we found frugivorous and insectivorous bats to differ significantly across the majority of kidney cell types by Chi-square tests of independence on Pearson residuals (**Extended Data Fig.6f**). All gene expression differences between bat kidney cell-types are reported in **Supplementary Table 6**. We have edited the text to provide more clarity on gene expression analysis:

“We conducted gene ontology (GO)^{9,10} enrichment analyses for the genes that showed differential expression between frugivorous and insectivorous bat kidneys (**Supplementary Table 6,7**).”

5. The authors converted bat peaks to human genome coordinates for downstream analysis. Over 60% bat peaks can align to human genome. For the other 30% bat-specific peak, it would be nice if the authors could further study the regulatory function, which will be important to explain the differences of glycometabolism between bat and human from an evolutionary point of view.

We agree with the reviewer that it would be nice if we could study the regulatory function of the 30% bat-specific peaks. While we are limited in available analysis tools with bat genomes, such as being unable to directly connect bat-specific peaks to genes, we attempted to address this suggestion by conducting differential motif enrichment with AME¹¹ to identify enriched motifs from scATAC-seq peaks from the bat-specific regions which did not align to the human reference genome.

Differentially enriched TFBSs from human-unmapped peaks and human-mapped were estimated with AME¹² using default parameters with JASPAR2022 CORE vertebrates as motif reference. We ranked enriched motifs based on their adjusted *p*-values and converted the ranks to percentile ranks in order to compare results with different total motif counts. In order to compare motif enrichments, we selected motifs that are ranked as the top 10% from each species and removed any motifs that are unique to one combination.

There was negative correlation between human-unmapped peaks vs. human-mapped peaks for each tissue of each species, indicating that the 30% bat-specific peaks have strong TFBS differences from the

60% bat peaks that align to the human genome (Fig. R3a-d). There was no correlation between human-unmapped peaks of each bat tissue (Fig. R3e-f), while there was positive correlation between human-mapped peaks of each bat tissue (Fig. R3g-h), suggesting that the TFBS in human-unmapped peaks largely differ between bats. The correlation between human-mapped peaks for bat kidney was slightly higher than the correlation between human-mapped peaks for bat pancreas, suggesting a higher degree of similarity for kidney compared to pancreas. In the regions of the bat genome that we have not explored, there may be more differences in pancreas. *GLI* motifs were more enriched in Jamaican fruit bats and *POU* motifs for big brown bats across both kidney and pancreas unmapped peaks for all combinations tested.

The major bias of this analysis is that peaks are declared “unmapped” by UCSC liftOver¹³ when they are actually found in duplicate regions of the human genome or “deleted” in the human genome. For this reason, we chose not to include this data in the manuscript.

E Insectivore Kidney unmapped vs Frugivore Kidney unmapped

insect_kid_unmapped: 192
fruit_kid_unmapped: 144
Overlap: 87

**F** Insectivore Pancreas unmapped vs Frugivore Pancreas unmapped

insect_pan_unmapped: 197
fruit_pan_unmapped: 180
Overlap: 96

**G** Insectivore Kidney mapped vs Frugivore Kidney mapped

insect_kid_mapped: 673
fruit_kid_mapped: 470
Overlap: 457

**H** Insectivore Pancreas mapped vs Frugivore Pancreas mapped

insect_pan_mapped: 708
fruit_pan_mapped: 450
Overlap: 436

Top 10% Motif for each sample

Figure R3: Motif enrichments of human-unmapped peaks and human-mapped peaks. In the scatter plots above, each point corresponds to a motif. Motifs that are ranked top 10% for each sample based on adjusted p-value are colored red. These motifs are also shown in the heatmaps below. Comparing fruit bat regions that map to the human genome with those that do not map in the (a) kidney and (b) pancreas. Comparing insectivorous bat regions that map to the human genome with those that do not map in the (c) kidney and (d) pancreas. Comparing fruit bat and insectivorous bat regions that do not map to the human genome in the (e) kidney and (f) pancreas. Comparing fruit bat and insectivorous bat regions that map to the human genome in the (g) kidney and (h) pancreas.

6. The authors indicated that fruit bats evolved many features similar to a human diabetic kidney with example of several motifs. Further system integration analysis between bat and human single-cell data is essential to improve the research significance of bat models.

We thank the reviewer for this suggestion and we have added several different comparisons to address it. We have added an evaluation of the homology and functional consistency of bat genes to human genes (**Extended Data Fig. 4a,b**) and pairwise synteny analyses (**Extended Data Fig. 4c**). Together, these results indicate that these bat genomes provide sufficient research significance since the vast majority of genes of interest are covered. Moreover, reference-based annotations in **Fig. 1i**, **Fig. 4g**, **Extended Data Fig. 6b** indicate conserved similarity between bat and human cell-types, since the method is only able to identify genes that exist in both human and bat references. Taken together, these analyses strengthen our findings that the bat kidney serves as a useful diabetes model.

7. In the gene regulatory part, the authors identified many TFs to explain the difference between species in kidney and pancreas. This is far from enough to understand the complex regulatory networks, especially for how these TFs regulate the different gene expression. A comprehensive multi-omics regulatory network needs to be constructed and further explored by bioinformatics tools such as Pando.

We thank the reviewer for this suggestion. We ran Pando separately for each species (with annotations from bats combined) and compared the results for each tissue. Pando generates gene regulatory networks (GRNs) based on both co-expression of genes and co-accessibility between peaks in those genes with common motifs. We have added the following text and figures to the manuscript:

Multi-omics gene regulatory network (GRN) analysis

Versions of insectivorous and fruit bat genomes were manually constructed for BSgenome¹⁴ (version 1.68.0), and these versions were used as reference genomes with multi-omic data into Pando (version 1.0.5)¹⁵ using default TF and TF motif lists built in Pando. We built GRNs separately for each species and each tissue. For inferring GRNs with `infer_grn()` function, we chose the “Signac” method for building peak-to-gene connections with variable genes identified for each species. To compare inferred GRNs between two species for the same organ, we calculated strength centrality with `igraph`¹⁶ (version 1.5.0) function “strength” and ranked each node with percentile ranks for each species. In order to select nodes that have both high centrality and specificity for each species, we selected nodes with top 100 or higher than top 50% centrality percentile ranks for one species and lower than top 25% centrality percentile ranks for the other species. With the species-specific top nodes, we ran pathway analysis with the following pathways: BioPlanet (2019)¹⁷, GO Biological Process (2023)^{9,10}, Reactome (2022)¹⁸, and WikiPathway Human (2021)¹⁹.

“To further understand gene regulatory differences between frugivorous and insectivorous bat kidneys, we conducted multi-omics gene regulatory network (GRN) analysis on renal epithelial cells with Pando¹⁵. To infer global GRNs, Pando simultaneously utilizes co-expression of TF genes and target genes and co-accessibility of TFBSs calculated from gene expression and chromatin accessibility profiles. Frugivore kidneys had GRNs enriched for acid-base and electrolyte homeostasis and glucose pathways, while insectivore kidneys had GRNs enriched for transcriptional and signal transduction pathways (**Extended Data Fig. 8a-c**). For example, *SLC13A2*, which encodes the sodium-dependent dicarboxylate transporter 1 (NaDC1), is a critical regulator of acid-base homeostasis and blood pressure and displays high centrality in the frugivore kidney, but not in the insectivore kidney (**Extended Data Fig. 8b**). *ADGRF5*, a regulator of acid-base homeostasis, displays high centrality in the insectivore kidney (**Extended Data Fig. 8b**). Adhesion-GPCR Gpr116 (*ADGRF5*) inhibits renal acid secretion by regulating vacuolar H⁺-ATPase expression²⁰, which may support the presence of “collecting duct acid secretion”, was found as an enriched pathway in frugivores (**Fig. 2d**) but not insectivores. We also observed notable differences within the networks of genes that have high strength centrality in both frugivores and insectivores. In the frugivore network of *NR3C2*, which encodes the mineralocorticoid receptor responsible for aldosterone-regulation of

electrolyte and blood pressure balance²¹, we found potassium transporter *KCNIP1* and vacuolar H⁺-ATPase subunits *ATP6V0A4* and *ATP6V1B1* to be upregulated and sulfate transporter *SLC13A1* to be downregulated (**Extended Data Fig. 8d**). We also observed insulin receptor substrate-1 (*IRS1*), *TBC1D4*, which regulates insulin-stimulated glucose uptake by controlling GLUT4 translocation²², and *TOX*, a transcriptional regulator in T1D²³ and associated with diabetic nephropathy²⁴, to be upregulated in our fruit bat. The major gluconeogenesis regulator *PCK1* and ketohexokinase gene *KHK*, which encodes the first enzyme to catabolize dietary fructose²⁵, were downregulated. In contrast, the *NR3C2* network in insectivore kidneys does not have these sugar regulation genes and instead involves different solute transporters than found in the fruit bat network. (**Extended Data Fig. 8d**). Together, these results indicate that the GRNs of frugivores and insectivores differ strongly in the kidney.”

“We next conducted multi-omics GRN analysis on endocrine and exocrine cell-types with Pando¹⁵. We found frugivore and insectivore pancreases to both have GRNs enriched for transcriptional and neuronal pathways (**Extended Data Fig. 12a-c**). Potassium channel gene *KCNB2* plays a role in regulating glucose-stimulated insulin secretion²⁶ and displays high centrality in frugivores, while potassium channel gene *KCNJ6*, which may also contribute to regulation of insulin secretion, displays high centrality in insectivores (**Extended Data Fig. 12b**). Upregulated in frugivore cells, *PTPRD*, which is involved in insulin signaling and associated with gestational diabetes risk^{27,28}, also displayed high centrality in frugivores (**Fig. 5c; Extended Data Fig. 12b**). Similar to our previous GRN kidney analysis, we observed differences within the networks of genes that have high strength centrality in both frugivores and insectivores. In the frugivore network of *ABCC8*, which was downregulated in frugivore beta cells (**Fig. 5c**), *ABCC8* was upregulated by *NEUROD1*, which both regulates endocrine cell maturation and function^{29,30}, and was downregulated by *GATA4*, which is required for pancreatic development and increases glucose-dependent insulinotropic polypeptide expression in a mouse beta cell line^{31,32} (**Extended Data Fig. 12d**). *ABCC8* was also downregulated by *GATA4* cofactor and insulin signaling regulator *ZFPM2*^{33,34}. The insectivore network of *ABCC8* involves fewer and different genes but shares *ABCC8* upregulation by *DACH1* and *RFX3*, both of which play roles in islet cell development and *RFX3* additionally regulates beta cell function and glucokinase expression^{35,36} (**Extended Data Fig. 12d**). Together, these results indicate that frugivores and insectivores share GRNs but the genes involved differ.”

More minor comments:

1. The legend titles of Fig.2a, Fig.3a and Fig.5d were missed.

Thanks for spotting this! We have added legend titles for Fig. 2a, Fig.3a, and Fig. 5d.

2. There are some issues with the formatting of reference index, for example, the 45.46 and 121.120 in the Methods.

Thank you for noticing. We fixed the formatting of all references.

Reviewer #2

Overall Comments:

In general, the authors have done a great deal of work, and they have used the state-of-the-art technology called scRNA-seq and scATAC-seq. Their findings are interesting, but require further validations. The authors must check the potential biases as follows:

1. The observed difference between the two bats may be caused by the difference of the two bat species per se rather than dietary difference. The best way to rule out this possibility is to detect difference between one species of Old World fruit bat and one species of its insectivore relative, and find the shared difference between the authors' comparison and my suggested new comparison.

The reviewer makes an important point that observed differences between the two bats in this study could be caused by the differences of the two bat species as opposed to their dietary differences, which we had noted in our discussion on limitations of this study, and following this comment, we have added to our discussion and revised the manuscript (abstract, introduction, and discussion) to be specific about our findings in Jamaican fruit bats. In comparing our data to available literature (page 12 paragraph 3, page 22 paragraph 1 and 2), it is unlikely that all of the observed differences are related to the species of bats as opposed to their dietary differences, which contribute to their species separation. We compared our gene expression findings to previous findings of convergent evolution between New World and Old World fruit bats from insectivorous bats³⁷⁻⁴¹, to a genome-wide investigation of genomic losses in OWFBs relative to insectivorous bats and other non-frugivorous mammals⁴², and between frugivorous bats and insectivorous bats⁴³⁻⁴⁶. However, there is also evidence for non-convergent evolution in New World and Old World fruit bats⁴⁷⁻⁴⁹, which we also used for comparison with our results.

2. The observed difference between the two bats may be caused by the difference in the ability of single-cell technology to capture expressed genes;

While there could be technical biases between the two bats for single-cell technology to capture expressed genes, more known factors that could cause such biases include differences in cell sizes from widely distinct cell populations, inherent RNA contents from different cell types, earlier versions of single cell platforms, and sample qualities^{50,51}. However, with use of matched tissue types and up-to-date single-cell platform, we minimized such technical biases. We have included summary statistics from cellranger-arc 2.0 (10x Genomics) in **Extended Data Fig. 1c & Supplementary Table 1**. For **Extended Data Fig. 1c**, we selected summary statistics that could indicate qualities of sequencing and library preparation and we observed consistency among different species. Furthermore, in order to reduce any unknown technical biases, we carefully selected quality control metrics from both RNA and ATAC profiles, performed up-to-date processing methods such as SCTransform for normalization and variance stabilization, and harmony combined with weighted nearest neighbor (WNN) approach for multi-omics profile integration.

3. The homology and functional consistency of cross-species genes need to be supported by sufficient data, especially the homology and functional consistency of protein-coding genes in the collinear regions of human and bat genomes needs to be proved.

The fruit bat genome used in this study (GCA_014825515.1) was annotated using homology-based prediction with the insect bat genome used in this study (GCF_000308155.1⁴¹). The insect bat protein alignments share > 76% average identity and > 79% average coverage with *Homo sapiens* known RefSeq of 36,324/37,337 sequences aligned by ProSplign (NCBI *Eptesicus fuscus* Annotation Release 100). The fruit bat protein alignments share > 77% average identity and > 83% average coverage with *Homo sapiens* known RefSeq of 44,707/60,894 sequences aligned by ProSplign (NCBI *Artibeus jamaicensis* Annotation Release 100).

The gene orthology across multiple bat species' has previously been explored using TOGA⁵², a machine learning model that leverages high quality reference gene annotations (such as those from mouse and human) and co-linear chain alignments between reference and query species to predict gene orthologs in the query species. For homology, TOGA can classify one-to-one, one-to-many, and many-to-many ortholog predictions. For functionality, TOGA classifies query species gene transcripts as intact, partially intact, missing, uncertain loss, lost, or paralogous. To reinforce the gene orthology and functionality supporting our work, we evaluated the TOGA predictions of fruit bat genome GCA_014825515.1 (*artJam2*) or insectivorous bat genome GCF_000308155.1 (*eptFus1*) bat gene orthologs using hg38 reference gene annotations. TOGA predicts the gene orthology at the transcript level, which supports the functional conservation of a gene.

TOGA one-to-one ortholog, transcript, and gene coordinate predictions were downloaded from https://genome.senckenberg.de/download/TOGA/human_hg38_reference/Chiroptera/ for *eptFus1* and *artJam2* (last downloaded July 1st, 2023). With differentially expressed genes we identified in bat kidneys and pancreas, we evaluated how many had at least one TOGA ortholog prediction between humans and the bat species. For transcript predictions, we asked how many of the differentially expressed genes were predicted to have at least one intact or partially intact gene transcript. For gene coordinates, we mapped predictions from the annotated chain to its paired scaffold.

To support pancreatic and kidney differentially expressed gene homology, we evaluated the frequency of one-to-one TOGA homolog predictions between humans and insectivorous (*eptFus1*) or frugivorous bats (*artJam2*). For differentially expressed pancreatic genes (N=4765; **Supplementary Table 10**), TOGA made ortholog predictions for 4650 human genes (98% of differentially expressed pancreatic genes). Among insectivores and frugivores, 4548 and 4547 genes were predicted to have human orthologs, respectively. Similarly among differentially expressed kidney genes (N=4132; **Supplementary Table 6**), TOGA made ortholog predictions for 4034 human genes (98% of differentially expressed kidney genes). In insectivores and frugivores, 3947 genes were predicted to have human orthologs. Together, this supports that our gene annotations are homologous and consistent between human and bat genomes.

To support the functional consistency of these TOGA ortholog predictions, we evaluated how often TOGA predicted differentially expressed ortholog transcripts were predicted to be intact or partially intact in frugivorous (ART) or insectivorous bats (EPT). We observe that between 95-96% of differentially expressed gene orthologs in human had at least 1 predicted intact/partially intact transcript in bats (see below). These TOGA predictions support that the majority of differentially expressed genes and their transcripts are predicted to have one-to-one orthologs and intact transcripts between humans and bats.

We have added the following text and figure to the manuscript: "We additionally evaluated the homology and functional consistency of cross-species genes that were differentially expressed between bats with inferred bat-human orthologs from TOGA⁵² (see **Methods**), finding that 95.5% and 95.6% of differentially expressed genes in bat kidney and pancreas, respectively, had a predicted ortholog in human (**Extended Data Fig. 4a**) and 95-96% of differentially expressed genes had a predicted intact or partially intact human orthologous transcript (**Extended Data Fig. 4b**)."

Specific comments:

1. Page 5, line 13 of paragraph 2. What are you comparing with? Why is there an increase of the number of shared features?

There is an increase of the number of shared features because some genes are named differently in each species annotation. For example, the gene *SEPTIN2* is also known as *SEPT2*. In the fruit bat annotation, this gene is named *SEPTIN2*, but in the insect bat annotation, the gene is named *SEPT2*. We used OrthoFinder to help us find genes of different names to re-name genes under common identifiers, the ID

numbers we report in Supplementary Table 16. OrthoFinder took proteomes from each bat species and created common identifiers for orthologous genes. To address this, we edited the text as follows:

“Genes often have multiple names, and names can vary across species annotations. Differing gene names affect integration by lowering the number of features shared between species. To improve integration of scRNA-seq and scATAC-seq between the two bat species, we used Orthofinder⁵³ to detect one-to-one orthologues, increasing the number of shared features by 3.11% across the big brown bat genome and by 3.24% across the Jamaican fruit bat genome.”

2. Page 6, line 12-13 of paragraph 1. As far as I know, kidney scRNA or scATAC in bats is not the first. Please see Lv et al., 2022 (<https://doi.org/10.1038/s41597-022-01447-7>)

Thank you for pointing this article out to us that we unfortunately missed. We have revised the text to read as follows:

“...our study presents the first joint functional genomics and gene expression dataset for bat kidney and pancreas,...”

We also compared our kidney cell-type identifications to the marker genes used in Lv et al. 2022: “In addition, we compared our kidney cell-type identifications to the insectivorous “horseshoe” bat *Rhinolophus affinis*⁸)...*LRP2*, *CUBN*, *SLC12A1*, *AQP2*, *PECAM1*, and *CD74* were also similarly used as marker genes for horseshoe bats⁸.”

3. Page 6, the end of paragraph 1. Authors need to provide the consistency of cell type classification between the scRNA/scATAC data of replicate treatments from the same tissue in the same bat species.

We have added **Extended Data Fig. 3**, which shows cell-type annotations across samples within each tissue of each species.

4. Page 9. “For example, ATPase H⁺ transporting V1 subunit G3 (*ATP6V1G3*) was highly expressed in type B intercalated cells in fruit bats but not in insectivorous bats.” Is this gene expressed in other tissues or cell types of insectivorous bats? It is necessary to exclude that it is not caused by the limitation that scRNA sequencing cannot comprehensively capture expressed genes?

In the methods we have the following text:

“For DEGs between species, we additionally filtered out genes that are not expressed in both species.”

We have added cellranger-arc 2.0 (10x Genomics) statistics in **Extended Data Fig. 1c** and **Supplementary Table 1**. *ATP6V1G3* is lowly expressed in other kidney cells and in pancreatic cells of insectivorous bats (Efus; see figures below). We have edited the manuscript accordingly:

“For example, ATPase H⁺ transporting V1 subunit G3 (*ATP6V1G3*) was highly expressed in type B intercalated cells in fruit bats and weakly in insectivorous bats (**Fig. 2e**).”

ATP6V1G3 Expression (Kidney)

ATP6V1G3 Expression (Pancreas)

Pancreas:

Figure R4: ATP6V1G3 expression in bat kidneys (top) and pancreases (bottom).

5. Page 13. "we converted our bat peaks to human genome coordinates (hg38), converting 141,779 insectivorous and 124,845 frugivorous bat peaks to hg38, respectively. We then used the Genomic Regions Enrichment of Annotations Tool (GREAT) to characterize the target genes associated with these peaks."Are they orthologous genes in the syntenic regions of bat and human genomes? In other words, how could you be sure that these human genes can be found in the syntenic regions of bat genomes?

We thank the reviewer for this comment. The chain files generated by UCSC Genome Bioinformatics Group for our insectivorous bat (<https://hgdownload.soe.ucsc.edu/goldenPath/hg38/vsEptFus1/>) and by us for our frugivorous bat (following the same methods) are alignments of the best/longest syntenic regions (http://genomewiki.ucsc.edu/index.php/LiftOver_Howto). Long regions of co-linearity are affected by our bat genomes being composed of many scaffolds, but for both bats, we only used peaks called in scaffolds > 50 kb (see **Bat genome modifications for joint scRNA and scATAC analysis**).

To further address this comment, we ran MCscan⁵⁴ to calculate pairwise synteny between each bat genome (scaffolds > 50 kb) and human and used mouse to human as a reference. While bat genomes are composed of scaffolds and have short scaffolds compared to chromosomal assemblies of mouse and human, the number of gene pairs for synteny blocks are similar among three species as well as when gene pair counts are normalized with the number of identified genes on the genome normalized. To be considered a syntenic block, the block must include 4 gene pairs (default). We found that increasing the threshold to 8 or 18 results in still equal values of normalized gene pair counts to that from mouse pairwise synteny. We have added the following figure and text to the manuscript: "We also evaluated pairwise syntenies of each bat and human using Mcscan⁵⁴ (see **Methods**) and found that the normalized number of gene pairs for synteny blocks for each bat pairwise synteny was similar to that of mouse and human pairwise synteny (**Extended Data Fig. 4c**)."

GREAT⁵⁵ advises using UCSC LiftOver utility to convert from non-model organism assemblies to the supported assemblies human or mouse, albeit with more caution (<https://great-help.atlassian.net/wiki/spaces/GREAT/pages/655414/Genome+Assemblies>). Following these guidelines, we use GREAT as a tool for broad and quick insights to understanding bat phenotypes in terms of human and mouse phenotypes, and we do not interpret the results beyond the report. We have edited the manuscript to reflect this (page 13, paragraph 2):

"We then used the Genomic Regions Enrichment of Annotations Tool (GREAT)⁵⁵ to identify the human target genes associated with these converted bat peaks."

6. Page 14. "Our cell-type-specific regulatory profilings allowed us to investigate human disease variants that are difficult to study without cell-type-specific data, such as rs79687284 at the PROX1 locus. This variant is highly associated with T2D and was predicted to have regulatory function as an enhancer, but it did not demonstrate transcriptional activity in luciferase reporter assays in a pancreatic beta cell line that expresses PROX1". I believe the cross-species analysis is particularly interesting as it highlights the potential for bats to serve as an alternative model for studying diabetes in humans. Through GWAS analysis, you were able to locate a candidate site in PROX1. Your results suggest that PROX1 performs opposite functions in insectivorous bats and humans. You have identified that PROX1 has an enhancer regulatory function associated with the rs79687284 variant. To further investigate this, you need to evaluate the degree of homology between PROX1 in bats and humans. Specifically, does the bat site corresponding to the rs79687284 site match? Additionally, it is important to determine if bat PROX1 also has the regulatory function of an enhancer. Furthermore, what are the findings of the scRNA and scATAC analysis of this gene in frugivorous bat kidneys?

We are grateful for these great comments about these findings. However, due to Reviewer 3's comment, who recommended we take out the GWAS section, we have removed all GWAS analysis from the manuscript.

Reviewer #3

Here, Gordon and colleagues perform a single nucleus multiomic comparison between fruit and insect bat kidneys and pancreas. Fruit bats have evolved to handle the higher sugar and potassium loads in their diets compared to insect bats but the precise cellular adaptations have not been characterized. The authors report increased collecting duct in fruit bats, which is consistent for important roles in potassium excretion and the fine tuning of sodium balance in this segment. In the pancreas, they report increased beta and alpha cells in fruit bats. They further find a diabetes-like signature in fruit bat proximal tubule including increased gluconeogenesis. A variety of individual genes and transcription factors and binding motif accessibility are analyzed with some validation by immunofluorescence as well.

Overall the study is well performed and reads clearly. The differing physiology is interesting. Limitations include the GWAS analysis and somewhat limited validation.

1. PT-like cells. These appear to be mostly characterized by decreased expression of PT anchor genes. Are there any upregulated genes compared to PT specific to this cluster? Single cell studies in human have identified other PT-like clusters with reduced expression of PT-defining genes, but also increased expression of others (*Vcam1*, *CD133*, *CCL2*, *Prom1*). Can the authors find any links between bat PT-like cells and these human PT cells that have variously been called “PT_VCAM1,” “Maladaptive PT” or “Failed-repair PT”?

We thank the reviewer for this suggestion. We had previously tried to annotate PT-like cells based on the reviewer’s suggested markers, but those markers, as well as *REL* and *RELA*⁵⁶, are currently not annotated in either bat genome. To address the reviewer’s suggestion, we have run DEG analysis on bat PT vs. bat PT-like cells to determine if there is shared gene expression between bat PT-like cells and human PT cell subtypes. We ran DEG analysis between bat PT and PT-like cells and reported it in **Supplementary Table 4**. As the reviewer notes, bat PT cells have higher expression of PT-defining genes. We did not find differential expression of *Havcr1*, *Mki67*, *Cdh13*, or *Kcnp4*⁵⁷ between bat PT and PT-like cells, but we found *Cald1* and *Pdgfd* to be upregulated and *Cdh6* to be downregulated in bat PT-like cells. Pathway analysis results (**Supplementary Table 4**) indicate that only bat PT cells are enriched for normal PT function, as has previously been reported⁵⁷. Bat PT-like cells are only enriched for *VEGFA-VEGFR2* Signaling (**Fig R5**), migration-activating pathway, and migration pathways were also enriched in injured PT cells^{56,57}. We have added the following text to the revised manuscript :

“This cluster may be synonymous to injured or regenerative proximal tubule cells in humans^{56,57} (**Supplementary Table 4**).”

Figure R5: Pathway enrichment analysis of DEG between bat PT and PT-like cells.

2. Does the GWAS analysis add anything? It does not make sense to me to take bat peaks and map human DM GWAS variants onto them, after lifting them over to the human genome. There is already plenty

of human snATAC-data published onto which the GWAS hits can be mapped. Doing so using the bat peaks would seem to be less sensitive. At the least, the authors need to compare bat vs human single cell ATAC datasets and prove that the bat analysis is providing new/better information.

Thank you for this comment. To address it, we removed all GWAS analysis from the revised manuscript.

3. In the discussion, the authors comment on increased Nfat5 binding motifs in insectivores and interpret this to be consistent with the increased protein diet in insectivores consume state, because Nfat5 has been shown to regulate proteinuria. But in fact podocyte Nfat5 activity causes proteinuria so would not be protective in the setting of a high protein diet.

We can't thank you enough for this comment. In reviewing our TFBS analyses, we, unfortunately, but gratefully, noticed that the enriched motifs were swapped for the wrong bat. *NFAT5* was in fact enriched in fruit bat kidneys, which makes sense given your comment. Moreover, we discovered a threshold error when calculating enrichment: motifs with $p \text{ adj} < e-100$ were being filtered out. In the revision, we fixed our motif enrichment calculations and have revised the following text to the Methods: "Differentially enriched TFBSs from cell-type-specific peaks were estimated with AME version 1.60¹², setting each time the other species' cell-type-specific peaks as control sequences and using default parameters with JASPAR2022 CORE vertebrates as motif reference. We ranked enriched motifs based on their adjusted p -values and converted the ranks to percentile ranks in order to compare results with different total motif counts. In order to compare motif enrichments, we selected motifs that are ranked as the top 10% from each species and cell type combination and removed any motifs that are unique to one combination."

We have re-analyzed all TFBS analyses appropriately in the manuscript:

"Next, to investigate differences in *cis*-regulation between frugivorous and insectivorous bats, we performed differential motif enrichment analyses on renal epithelial cells using Analysis of Motif Enrichment (AME¹²). For collecting duct cells, we found two clusters of differentially enriched motifs, which separated frugivorous and insectivorous bats (**Fig. 3a**). Frugivorous bat collecting duct cells were enriched for *NKX*, *FOX*, and *NFAT* TFBS (**Fig. 3a**). *NKX* and *FOX* TFs are broadly expressed and important for development. *NFATc* and *NFAT5* are regulated by tonicity and protect the kidney from osmotic stress by activating genes that contribute to cellular accumulation of organic osmolytes, such as *AQP2*^{58,59}. The high water intake frugivores get from fruit may contribute to higher osmotic stress in the kidney, and *AQP2* was upregulated in frugivore principal cells (**Extended Data Fig. 7b**). Insectivorous bat collecting duct cells were enriched for *E2F* and *EGR* motifs (**Fig. 3a**). *E2F* TFs are involved in cell cycle regulation⁶⁰. *EGR1* knockdown may alleviate renal injury in diabetic kidney disease (DKD) by downregulating renin and RAAS⁶¹. Insectivorous bat collecting duct cells were also enriched for *KLF15* (**Fig. 3a**), which protects against podocyte injury⁶², and *TFAP2A*, which maintains adult mouse medullary collecting duct structure⁶³. Consistent with the medullary composition differences between bats, insectivorous bats have upregulated *TFAP2A* expression in principal cells (**Extended Data Fig. 7g**). Collectively, differential motif enrichment analyses on bat collecting duct cells suggest the importance of osmoregulation in fruit bats and of medulla maintenance in insectivorous bats.

We also separated proximal tubules, proximal tubule-like cells, distal convoluted tubules, TAL and DTL for differential motif enrichment and found multiple TFBS clusters that differentiate frugivorous and insectivorous bats, including many of the same TFs identified in collecting duct cells (**Fig. 3a**). High glucose-induced inactivation of *FOXO1* likely contributes to DKD pathogenesis⁶⁴, and *FOXO1* was downregulated in fruit bat distal convoluted tubules, connecting tubules, TAL and DTL (**Extended Data Fig. 7g**). Enriched in fruit bat tubules, *FOXO3* protects against kidney injury in T2D nephropathy⁶⁵, and the *FOXO3* motif was also enriched in human diabetic proximal tubules⁶⁶. Fruit bat tubules also shared enrichment with collecting duct cells for vitamin D receptor (*VDR*) motifs (**Fig. 3a**). *VDR* is highly expressed in renal tubules and plays a renoprotective role in a diabetic mouse model⁶⁷. *SOX* TFs were more enriched in fruit bat tubules than in collecting duct cells (**Fig. 3a**). *SOX6* is upregulated after a low sodium diet in mouse and controls renin secretion⁶⁸. In addition to *TFAP2A* motifs, insectivore tubules were enriched for

TFAP2B motifs (**Fig. 3a**). *TFAP2B* is critical for the formation and function of distal convoluted tubules⁶³ and was upregulated in insectivore distal convoluted tubules (**Extended Data Fig. 7g**). Insectivore tubules were notably enriched for *POU* TFBS, some of which were shared with frugivore TAL (**Fig. 3a**). *POU* TFs have multiple roles in development and neuroendocrine function⁶⁹. In summary, kidney TFBS differential enrichment between insectivorous and frugivorous bats identified key TFs involved in diet, with frugivorous bats demonstrating some diabetic-associated motif signatures.”

“We then used AME¹² to identify differentially enriched TFBS in scATAC-seq peaks between insectivorous and frugivorous bats in endocrine and exocrine cell-types. We found that there are two clusters of TFBSs that show differential enrichment across alpha and beta cells of the two bats (**Fig. 5d**). Similar to renal epithelial cells, *FOX* motifs, such as *FOXP1*, were enriched in fruit bat endocrine cells (**Fig. 5d**). *FOXP1* is required for alpha cell proliferation and function⁷⁰, and *FOXP1* is upregulated in fruit bat alpha cells as well as beta cells (**Extended Data Fig. 11f**). Fruit bat endocrine cells were also strongly enriched for *HOX* gene TFBS (**Fig. 5d**), which play roles in pancreatic development⁷¹. TFBS for *PDX1*, which regulates endocrine and exocrine cell development and binds the mouse insulin enhancer⁷², were notably enriched in fruit bat endocrine cells (**Fig. 5d**). Insectivorous bat endocrine cells were enriched for *SP* and *KLF* motifs (**Fig. 5d**), TFs that regulate the pathophysiology of the digestive system, many of which maintain beta cell function and control oxidative stress response genes⁷³. Similar to the kidney, insectivorous bat endocrine cells were also enriched for *E2F* motifs, including *E2F1* (**Fig. 5d**), which regulates glucagon-like peptide-1 during insulin secretion⁷⁴, and *EGR1*, which regulates glucose homeostasis and pancreatic islet size⁷⁵. Together, TFBS analyses demonstrate that the TF families utilized in endocrine pancreas regulation strongly differ between bats.

Two clusters of TFBSs were also annotated for exocrine cell-types, indicating again that there is a divergence in TFBS usage between these two bat species (**Fig. 5d**). In addition to endocrine cells, fruit bat exocrine cells were enriched for *FOX* TFBS (**Fig. 5d**). Fruit bat exocrine cells were also enriched for *NFAT* motifs, which were enriched in the fruit bat kidney (**Fig. 3a**). *NFATc1-3* are expressed in acinar cells and promote pancreatic growth⁷⁶. *GATA4*, that is expressed in acinar cells and controls mouse pancreas organogenesis, was also enriched in fruit bat exocrine cells (**Fig. 5d**). Similar to fruit bat renal tubules, *SOX* motifs, which are necessary for developmental and physiological regulation of the pancreas⁷⁷, were highly enriched in fruit bat exocrine cells (**Fig. 3a, 5d**), including *SOX11*, which *SOX11*-deficient mouse embryos exhibit pancreatic hypoplasia⁷⁷. In addition to endocrine cells, insectivore exocrine cells were enriched for *SP* and *KLF* TFBS (**Fig. 5d**). *SP1* maintains acinar cell function and inhibits proliferation, and both *SP1* and *SP3* regulate the expression of the secretin receptor gene⁷³. Insectivore exocrine cells were also enriched for *TFAP* TFBS (**Fig. 5d**), which were also identified in the kidney (**Fig. 3a**). *TFAP2B* was identified as an obesity-associated loci⁷⁸. Taken together, these bat species differ strongly in TF usage in the pancreas.”

References

1. Stuart, T., Srivastava, A., Madad, S., Lareau, C. A. & Satija, R. Single-cell chromatin state analysis with Signac. *Nat. Methods* **18**, 1333–1341 (2021).
2. Hafemeister, C. & Satija, R. Normalization and variance stabilization of single-cell RNA-seq data using regularized negative binomial regression. *Genome Biol.* **20**, 296 (2019).
3. Young, M. D. & Behjati, S. SoupX removes ambient RNA contamination from droplet-based single-cell RNA sequencing data. *Gigascience* **9**, (2020).
4. Wolock, S. L., Lopez, R. & Klein, A. M. Scrublet: Computational Identification of Cell Doublets in Single-Cell Transcriptomic Data. *Cell Syst* **8**, 281–291.e9 (2019).
5. Hao, Y. *et al.* Integrated analysis of multimodal single-cell data. *Cell* **184**, 3573–3587.e29 (2021).
6. Lake, B. B. *et al.* An atlas of healthy and injured cell states and niches in the human kidney. *bioRxiv* 2021.07.28.454201 (2021) doi:10.1101/2021.07.28.454201.
7. Tosti, L. *et al.* Single-Nucleus and In Situ RNA–Sequencing Reveal Cell Topographies in the Human Pancreas. *Gastroenterology* **160**, 1330–1344.e11 (2021).
8. Lv, T. *et al.* A map of bat virus receptors derived from single-cell multiomics. *Sci Data* **9**, 336 (2022).
9. Ashburner, M. *et al.* Gene ontology: tool for the unification of biology. The Gene Ontology Consortium. *Nat. Genet.* **25**, 25–29 (2000).
10. Gene Ontology Consortium *et al.* The Gene Ontology knowledgebase in 2023. *Genetics* **224**, (2023).
11. Bailey, T. L., Johnson, J., Grant, C. E. & Noble, W. S. The MEME Suite. *Nucleic Acids Res.* **43**, W39–49 (2015).
12. McLeay, R. C. & Bailey, T. L. Motif Enrichment Analysis: a unified framework and an evaluation on ChIP data. *BMC Bioinformatics* **11**, 165 (2010).
13. Hinrichs, A. S. *et al.* The UCSC Genome Browser Database: update 2006. *Nucleic Acids Res.* **34**, D590–8 (2006).
14. Pagès, H. *BSgenome: Software infrastructure for efficient representation of full genomes and their SNPs.* (2023).
15. Fleck, J. S. *et al.* Inferring and perturbing cell fate regulomes in human brain organoids. *Nature* 1–8 (2022).
16. Csárdi, G. *et al.* *igraph for R: R interface of the igraph library for graph theory and network analysis.* (2023). doi:10.5281/zenodo.8046777.

17. Huang, R. *et al.* The NCATS BioPlanet - An Integrated Platform for Exploring the Universe of Cellular Signaling Pathways for Toxicology, Systems Biology, and Chemical Genomics. *Front. Pharmacol.* **10**, 445 (2019).
18. Gillespie, M. *et al.* The reactome pathway knowledgebase 2022. *Nucleic Acids Res.* **50**, D687–D692 (2022).
19. Martens, M. *et al.* WikiPathways: connecting communities. *Nucleic Acids Res.* **49**, D613–D621 (2021).
20. Zaidman, N. A. *et al.* Adhesion-GPCR Gpr116 (ADGRF5) expression inhibits renal acid secretion. *Proc. Natl. Acad. Sci. U. S. A.* **117**, 26470–26481 (2020).
21. Jung, H. J., Su, X.-T., Al-Qusairi, L., Ellison, D. H. & Welling, P. A. Architecture of the distal nephron mineralocorticoid receptor-dependent transcriptome defined. *FASEB J.* **34**, 1–1 (2020).
22. Binsch, C. *et al.* Deletion of Tbc1d4/As160 abrogates cardiac glucose uptake and increases myocardial damage after ischemia/reperfusion. *Cardiovasc. Diabetol.* **22**, 17 (2023).
23. Albanus, R. D. *et al.* Single-cell gene expression and chromatin accessibility profiling of human pancreatic islets at basal and stimulatory conditions nominates mechanisms of type 1 diabetes genetic risk. *bioRxiv* 2022.11.12.516291 (2022) doi:10.1101/2022.11.12.516291.
24. Wei, F. *et al.* TOX and CDKN2A/B Gene Polymorphisms Are Associated with Type 2 Diabetes in Han Chinese. *Sci. Rep.* **5**, 11900 (2015).
25. Kim, J. *et al.* Kethexokinase-A acts as a nuclear protein kinase that mediates fructose-induced metastasis in breast cancer. *Nat. Commun.* **11**, 5436 (2020).
26. Jensen, M. V. *et al.* Control of voltage-gated potassium channel Kv2.2 expression by pyruvate-isocitrate cycling regulates glucose-stimulated insulin secretion. *J. Biol. Chem.* **288**, 23128–23140 (2013).
27. Chen, Y.-T. *et al.* PTPRD silencing by DNA hypermethylation decreases insulin receptor signaling and leads to type 2 diabetes. *Oncotarget* **6**, 12997–13005 (2015).
28. Kang, Y., Huang, H., Li, H., Sun, W. & Zhang, C. Functional genetic variants in the 3'UTR of PTPRD associated with the risk of gestational diabetes mellitus. *Exp. Ther. Med.* **21**, 562 (2021).
29. Naya, F. J., Stellrecht, C. M. & Tsai, M. J. Tissue-specific regulation of the insulin gene by a novel basic helix-loop-helix transcription factor. *Genes Dev.* **9**, 1009–1019 (1995).
30. Bohuslavova, R. *et al.* NEUROD1 Is Required for the Early α and β Endocrine Differentiation in the Pancreas. *Int. J. Mol. Sci.* **22**, (2021).

31. Jepeal, L. I., Boylan, M. O. & Michael Wolfe, M. GATA-4 upregulates glucose-dependent insulinotropic polypeptide expression in cells of pancreatic and intestinal lineage. *Mol. Cell. Endocrinol.* **287**, 20–29 (2008).
32. Ding, L. *et al.* Identification and functional study of GATA4 gene regulatory variants in type 2 diabetes mellitus. *BMC Endocr. Disord.* **21**, 73 (2021).
33. Lu, J. R. *et al.* FOG-2, a heart- and brain-enriched cofactor for GATA transcription factors. *Mol. Cell. Biol.* **19**, 4495–4502 (1999).
34. Hyun, S. *et al.* Conserved MicroRNA miR-8/miR-200 and its target USH/FOG2 control growth by regulating PI3K. *Cell* **139**, 1096–1108 (2009).
35. Ait-Lounis, A. *et al.* The transcription factor Rfx3 regulates beta-cell differentiation, function, and glucokinase expression. *Diabetes* **59**, 1674–1685 (2010).
36. Kalousova, A. *et al.* Dachshund homologues play a conserved role in islet cell development. *Dev. Biol.* **348**, 143–152 (2010).
37. Liu, Y., Xu, H., Yuan, X., Rossiter, S. J. & Zhang, S. Multiple adaptive losses of alanine-glyoxylate aminotransferase mitochondrial targeting in fruit-eating bats. *Mol. Biol. Evol.* **29**, 1507–1511 (2012).
38. Qian, Y., Fang, T., Shen, B. & Zhang, S. The glycogen synthase 2 gene (*Gys2*) displays parallel evolution between Old World and New World fruit bats. *J. Mol. Evol.* **78**, 66–74 (2014).
39. Fang, L., Shen, B., Irwin, D. M. & Zhang, S. Parallel evolution of the glycogen synthase 1 (muscle) gene *Gys1* between Old World and New World fruit bats (Order: Chiroptera). *Biochem. Genet.* **52**, 443–458 (2014).
40. Meng, F., Zhu, L., Huang, W., Irwin, D. M. & Zhang, S. Bats: Body mass index, forearm mass index, blood glucose levels and *SLC2A2* genes for diabetes. *Sci. Rep.* **6**, 29960 (2016).
41. Wang, K. *et al.* Molecular adaptation and convergent evolution of frugivory in Old World and neotropical fruit bats. *Mol. Ecol.* **29**, 4366–4381 (2020).
42. Sharma, V. *et al.* A genomics approach reveals insights into the importance of gene losses for mammalian adaptations. *Nat. Commun.* **9**, 1215 (2018).
43. Schondube, J. E., Herrera-M, L. G. & Martínez del Rio, C. Diet and the evolution of digestion and renal function in phyllostomid bats. *Zoology* **104**, 59–73 (2001).
44. Casotti, G., Gerardo Herrera M, L., Flores M, J. J., Mancina, C. A. & Braun, E. J. Relationships between renal morphology and diet in 26 species of new world bats (suborder microchiroptera).

- Zoology* **109**, 196–207 (2006).
45. Machado-Santos, C. *et al.* Influence of feeding habits in the endocrine pancreas of insectivore bat *Pteronotus personatus* and nectarivore bat *Anoura geoffroyi*: A comparative stereological and immunohistochemical study. *Tissue Cell* **49**, 1–7 (2017).
 46. Moreno-Santillán, D. D., Machain-Williams, C., Hernández-Montes, G. & Ortega, J. De Novo Transcriptome Assembly and Functional Annotation in Five Species of Bats. *Sci. Rep.* **9**, 6222 (2019).
 47. Shen, B., Han, X., Zhang, J., Rossiter, S. J. & Zhang, S. Adaptive evolution in the glucose transporter 4 gene *Slc2a4* in Old World fruit bats (family: Pteropodidae). *PLoS One* **7**, e33197 (2012).
 48. Shen, B. *et al.* Relaxed evolution in the tyrosine aminotransferase gene *tat* in old world fruit bats (Chiroptera: Pteropodidae). *PLoS One* **9**, e97483 (2014).
 49. Yin, Q. *et al.* Molecular Evolution of the Nuclear Factor (Erythroid-Derived 2)-Like 2 Gene *Nrf2* in Old World Fruit Bats (Chiroptera: Pteropodidae). *PLoS One* **11**, e0146274 (2016).
 50. Mereu, E. *et al.* Benchmarking single-cell RNA-sequencing protocols for cell atlas projects. *Nat. Biotechnol.* **38**, 747–755 (2020).
 51. Yamawaki, T. M. *et al.* Systematic comparison of high-throughput single-cell RNA-seq methods for immune cell profiling. *BMC Genomics* **22**, 66 (2021).
 52. Kirilenko, B. M. *et al.* Integrating gene annotation with orthology inference at scale. *Science* **380**, eabn3107 (2023).
 53. Emms, D. M. & Kelly, S. OrthoFinder: phylogenetic orthology inference for comparative genomics. *Genome Biol.* **20**, 238 (2019).
 54. Tang, H. *et al.* Synteny and collinearity in plant genomes. *Science* **320**, 486–488 (2008).
 55. McLean, C. Y. *et al.* GREAT improves functional interpretation of cis-regulatory regions. *Nat. Biotechnol.* **28**, 495–501 (2010).
 56. Muto, Y. *et al.* Single cell transcriptional and chromatin accessibility profiling redefine cellular heterogeneity in the adult human kidney. *Nat. Commun.* **12**, 2190 (2021).
 57. Gerhardt, L. M. S., Liu, J., Koppitch, K., Cippà, P. E. & McMahon, A. P. Single-nuclear transcriptomics reveals diversity of proximal tubule cell states in a dynamic response to acute kidney injury. *Proc. Natl. Acad. Sci. U. S. A.* **118**, (2021).
 58. Burg, M. B., Ferraris, J. D. & Dmitrieva, N. I. Cellular response to hyperosmotic stresses. *Physiol. Rev.* **87**, 1441–1474 (2007).

59. Li, S.-Z. *et al.* Calcineurin-NFATc signaling pathway regulates AQP2 expression in response to calcium signals and osmotic stress. *Am. J. Physiol. Cell Physiol.* **292**, C1606–16 (2007).
60. Attwooll, C., Lazzerini Denchi, E. & Helin, K. The E2F family: specific functions and overlapping interests. *EMBO J.* **23**, 4709–4716 (2004).
61. Hu, F. *et al.* Egr1 Knockdown Combined with an ACE Inhibitor Ameliorates Diabetic Kidney Disease in Mice: Blockade of Compensatory Renin Increase. *Diabetes Metab. Syndr. Obes.* **13**, 1005–1013 (2020).
62. Rane, M. J., Zhao, Y. & Cai, L. Krüppel-like factors (KLFs) in renal physiology and disease. *EBioMedicine* **40**, 743–750 (2019).
63. Lamontagne, J. O. *et al.* Transcription factors AP-2 α and AP-2 β regulate distinct segments of the distal nephron in the mammalian kidney. *Nat. Commun.* **13**, 2226 (2022).
64. Wang, Y. & He, W. Improving the Dysregulation of FoxO1 Activity Is a Potential Therapy for Alleviating Diabetic Kidney Disease. *Front. Pharmacol.* **12**, 630617 (2021).
65. Wang, X., Ji, T., Li, X., Qu, X. & Bai, S. FOXO3a Protects against Kidney Injury in Type II Diabetic Nephropathy by Promoting Sirt6 Expression and Inhibiting Smad3 Acetylation. *Oxid. Med. Cell. Longev.* **2021**, 5565761 (2021).
66. Wilson, P. C. *et al.* Multimodal single cell sequencing implicates chromatin accessibility and genetic background in diabetic kidney disease progression. *Nat. Commun.* **13**, 5253 (2022).
67. Li, A. *et al.* Vitamin D-VDR (vitamin D receptor) regulates defective autophagy in renal tubular epithelial cell in streptozotocin-induced diabetic mice via the AMPK pathway. *Autophagy* **18**, 877–890 (2022).
68. Saleem, M. *et al.* Sox6 as a new modulator of renin expression in the kidney. *Am. J. Physiol. Renal Physiol.* **318**, F285–F297 (2020).
69. Andersen, B. & Rosenfeld, M. G. POU domain factors in the neuroendocrine system: lessons from developmental biology provide insights into human disease. *Endocr. Rev.* **22**, 2–35 (2001).
70. Spaeth, J. M. *et al.* The FOXP1, FOXP2 and FOXP4 transcription factors are required for islet alpha cell proliferation and function in mice. *Diabetologia* **58**, 1836–1844 (2015).
71. Kuo, T.-L., Cheng, K.-H., Chen, L.-T. & Hung, W.-C. Deciphering The Potential Role of Hox Genes in Pancreatic Cancer. *Cancers* **11**, (2019).
72. Zhu, Y., Liu, Q., Zhou, Z. & Ikeda, Y. PDX1, Neurogenin-3, and MAFA: critical transcription regulators for beta cell development and regeneration. *Stem Cell Res. Ther.* **8**, 240 (2017).

73. Kim, C.-K., He, P., Bialkowska, A. B. & Yang, V. W. SP and KLF Transcription Factors in Digestive Physiology and Diseases. *Gastroenterology* **152**, 1845–1875 (2017).
74. Bourouh, C. *et al.* The transcription factor E2F1 controls the GLP-1 receptor pathway in pancreatic β cells. *Cell Rep.* **40**, 111170 (2022).
75. Thiel, G. & Rössler, O. G. Glucose Homeostasis and Pancreatic Islet Size Are Regulated by the Transcription Factors Elk-1 and Egr-1 and the Protein Phosphatase Calcineurin. *Int. J. Mol. Sci.* **24**, (2023).
76. Gurda, G. T., Guo, L., Lee, S.-H., Molkentin, J. D. & Williams, J. A. Cholecystokinin activates pancreatic calcineurin-NFAT signaling in vitro and in vivo. *Mol. Biol. Cell* **19**, 198–206 (2008).
77. Yin, C. Molecular mechanisms of Sox transcription factors during the development of liver, bile duct, and pancreas. *Semin. Cell Dev. Biol.* **63**, 68–78 (2017).
78. Williams, M. J. *et al.* Regulation of aggression by obesity-linked genes TfAP-2 and Twz through octopamine signaling in *Drosophila*. *Genetics* **196**, 349–362 (2014).

REVIEWERS' COMMENTS

Reviewer #1 (Remarks to the Author):

The authors have addressed the concerns brought up in the first round. This is a highly responsive revision and this work will be of great utility to the scientific community. I congratulate the team for this important contribution.

Reviewer #2 (Remarks to the Author):

The manuscript has undergone significant improvements in methodology and reliability of results after revision, although the authors are still unable to solve the following issue - the observed differences between the two bats in this study could be caused by the differences of the two bat species as opposed to their dietary differences. If this work is to be published, I have three additional comments as follows:

1) It is strongly recommended that the authors provide information on the gene orthologs between bats and humans for the main results highlighted in the paper. This should include details such as the gene name, identity, and coverage ratios. This step is crucial to ensure that major genetic findings are not misinterpreted due to limitations in comparing bat genomes. Additionally, including this information will enhance the readers' assessment of the consistency of bats as a model for human diabetes research at the gene level.

2) It is suggested that the sentence "This was apparent in our results with the insectivorous bat genome having more annotated genes shared with humans than the fruit bat genome" be removed from the discussion section. This recommendation is made because the continuity of the two bat genomes is relatively low, indicated by the short Contig N50, which may result in multiple uncertain assemblies of gene sequences.

3) It is worth verifying the default parameters of MCscan, as it is commonly known that the number of genes required to call a collinear block is 5 rather than 4. Please double-check this parameter to ensure accuracy.

Reviewer #3 (Remarks to the Author):

The authors have adequately addressed my concerns, and the new Nfat5 re-analysis is reassuring.

We thank the reviewers for their positive assessment of our revisions and for their approval of our significantly improved manuscript. Below is a point-by-point response to these comments with our answers in blue font.

Reviewer #1 (Remarks to the Author):

The authors have addressed the concerns brought up in the first round. This is a highly responsive revision and this work will be of great utility to the scientific community. I congratulate the team for this important contribution.

We thank you very much for your time and great comments.

Reviewer #2 (Remarks to the Author):

The manuscript has undergone significant improvements in methodology and reliability of results after revision, although the authors are still unable to solve the following issue - the observed differences between the two bats in this study could be caused by the differences of the two bat species as opposed to their dietary differences. If this work is to be published, I have three additional comments as follows:

1) It is strongly recommended that the authors provide information on the gene orthologs between bats and humans for the main results highlighted in the paper. This should include details such as the gene name, identity, and coverage ratios. This step is crucial to ensure that major genetic findings are not misinterpreted due to limitations in comparing bat genomes. Additionally, including this information will enhance the readers' assessment of the consistency of bats as a model for human diabetes research at the gene level.

We have revised Supplementary Table 14 to include GeneIDs. OrthoFinder does not define orthologs by coverage as sequence similarity in the absence of phylogeny is estimative. OrthoFinder conducts phylogenetic reconstruction of all species' genes using proteomes as input to appropriately define orthologs(Emms and Kelly 2019).

2) It is suggested that the sentence "This was apparent in our results with the insectivorous bat genome having more annotated genes shared with humans than the fruit bat genome" be removed from the discussion section. This recommendation is made because the continuity of the two bat genomes is relatively low, indicated by the short Contig N50, which may result in multiple uncertain assemblies of gene sequences.

We have removed this sentence from the discussion.

3) It is worth verifying the default parameters of MCscan, as it is commonly known that the number of genes required to call a collinear block is 5 rather than 4. Please double-check this parameter to ensure accuracy.

We performed MCscan with default parameters, which define the number of genes to be 4.

Reviewer #3 (Remarks to the Author):

The authors have adequately addressed my concerns, and the new Nfat5 re-analysis is reassuring.

We thank the reviewer again for pointing out the first Nfat5 analysis, which allowed us to appropriately re-analyze.